# TimeKAN: KAN-based Frequency Decomposition Learning Architecture for Long-term Time Series Forecasting

**Songtao Huang**[1,2]**, Zhen Zhao**[1]**, Can Li**[3]**, Lei Bai**[1✉]
[1]Shanghai Artificial Intelligence Laboratory, Shanghai, China
[2]School of Information Science and Engineering, Lanzhou University, Lanzhou, China
[3]The Key Laboratory of Road and Traffic Engineering of the Ministry of Education,
  Tongji University, Shanghai, China
`huangsongtao@pjlab.org.cn`, `zhen.zhao@outlook.com`,
`lchelen1005@gmail.com`, `baisanshi@gmail.com`

## Abstract

Real-world time series often have multiple frequency components that are intertwined with each other, making accurate time series forecasting challenging. Decomposing the mixed frequency components into multiple single frequency components is a natural choice. However, the information density of patterns varies across different frequencies, and employing a uniform modeling approach for different frequency components can lead to inaccurate characterization. To address this challenges, inspired by the flexibility of the recent Kolmogorov-Arnold Network (KAN), we propose a KAN-based Frequency Decomposition Learning architecture (TimeKAN) to address the complex forecasting challenges caused by multiple frequency mixtures. Specifically, TimeKAN mainly consists of three components: Cascaded Frequency Decomposition (CFD) blocks, Multi-order KAN Representation Learning (M-KAN) blocks and Frequency Mixing blocks. CFD blocks adopt a bottom-up cascading approach to obtain series representations for each frequency band. Benefiting from the high flexibility of KAN, we design a novel M-KAN block to learn and represent specific temporal patterns within each frequency band. Finally, Frequency Mixing blocks is used to recombine the frequency bands into the original format. Extensive experimental results across multiple real-world time series datasets demonstrate that TimeKAN achieves state-of-the-art performance as an extremely lightweight architecture. Code is available at `https://github.com/huangst21/TimeKAN`.

## 1 Introduction

Time series forecasting (TSF) has garnered significant interest due to its wide range of applications, including finance (Huang et al., 2024), energy management (Yin et al., 2023), traffic flow planning (Jiang & Luo, 2022), and weather forecasting (Lam et al., 2023). Recently, deep learning has led to substantial advancements in TSF, with the most state-of-the-art performances achieved by CNN-based methods (Wang et al., 2023; donghao & wang xue, 2024), Transformer-based methods(Nie et al., 2023; Liu et al., 2024b) and MLP-based methods (Zeng et al., 2023; Wang et al., 2024a).

Due to the complex nature of the real world, observed multivariate time series are often non-stationary and exhibit diverse patterns. These intertwined patterns complicate the internal relationships within the time series, making it challenging to capture and establish connections between historical observations and future targets. To address the complex temporal patterns in time series, an increasing number of studies focus on leveraging prior knowledge to decompose time series into simpler components that provide a basis for forecasting. For instance, Autoformer (Wu et al., 2021) decomposes time series into seasonal and trend components. This idea is also adopted by DLinear (Zeng et al., 2023) and FEDFormer (Zhou et al., 2022b). Building on this foundation, TimeMixer (Wang et al., 2024a) further introduces multi-scale seasonal-trend decomposition and highlights the importance of interactions between different scales. Recent models like TimesNet (Wu et al.,

2023), PDF (Dai et al., 2024), and SparseTSF (Lin et al., 2024) emphasize the inherent periodicity in time series and decompose long sequences into multiple shorter ones based on the period length, thereby enabling the separate modeling of inter-period and intra-period dependencies within temporal patterns. In summary, these different decomposition methods share a common goal: utilizing the simplified subsequences to provide critical information for future predictions, thereby achieving accurate forecasting.

It is worth noting that time series are often composed of multiple frequency components, where the low-frequency components represent long-term periodic variations and the high-frequency components capture certain abrupt events. The mixture of different frequency components makes accurate forecasting particularly challenging. The aforementioned decomposition approaches motivate us to design a frequency decomposition framework that decouples different frequency components in a time series and independently learns the temporal patterns associated with each frequency. However, this introduces another challenge: the information density of patterns varies across different frequencies, and employing a uniform modeling approach for different frequency components can lead to inaccurate characterizations, resulting in sub-optimal results. Fortunately, a new neural network architecture, known as Kolmogorov-Arnold Networks (KAN) (Liu et al., 2024c), has recently gained significant attention in the deep learning community due to its outstanding data-fitting capabilities and flexibility, showing potential as a substitute for traditional MLP. Compared to MLP, KAN offers optional kernels and allows for the adjustment of kernel order to control its fitting capacity. This consideration leads us to explore the use of Multi-order KANs to represent temporal patterns across different frequencies, thereby providing more accurate information for forecasting.

Motivated by these observations, we propose a KAN-based Frequency Decomposition Learning architecture (TimeKAN) to address the complex prediction challenges caused by multiple frequency mixtures. Specifically, TimeKAN first employs moving average to progressively remove relatively high-frequency components from the sequence. Subsequently, Cascaded Frequency Decomposition (CFD) blocks adopt a bottom-up cascading approach to obtain sequence representations for each frequency band. Multi-order KAN Representation Learning (M-KAN) blocks leverage the high flexibility of KAN to learn and represent specific temporal patterns within each frequency band. Finally, Frequency Mixing blocks recombine the frequency bands into the original format, ensuring that this Decomposition-Learning-Mixing process is repeatable, thereby modeling different temporal patterns at various frequencies more accurately. The final high-level sequence is then mapped to the desired forecasting output via a simple linear mapping. With our meticulously designed architecture, TimeKAN achieves state-of-the-art performance across multiple long-term time series forecasting tasks, while also being a lightweight architecture that outperforms complex TSF models with fewer computational resources.

Our contributions are summarized as follows:

- We revisit time series forecasting from the perspective of frequency decoupling, effectively disentangling time series characteristics through a frequency Decomposition-Learning-Mixing architecture to address challenges caused by complex information coupling in time series.

- We introduce TimeKAN as a lightweight yet effective forecasting model and design a novel M-KAN blocks to effectively modeling and representing patterns at different frequencies by maximizing the flexibility of KAN.

- TimeKAN demonstrates superior performance across multiple TSF prediction tasks, while having a parameter count significantly lower than that of state-of-the-art TSF models.

## 2 RELATED WORK

### 2.1 KOLMOGOROV-ARNOLD NETWORK

Kolmogorov-Arnold representation theorem states that any multivariate continuous function can be expressed as a combination of univariate functions and addition operations. Kolmogorov-Arnold Network (KAN) (Liu et al., 2024c) leverages this theorem to propose an innovative alternative to traditional MLP. Unlike MLP, which use fixed activation functions at the nodes, KAN introduces

learnable activation functions along the edges. Due to the flexibility and adaptability, KAN is considered as a promising alternative to MLP.

The original KAN was parameterized using spline functions. However, due to the inherent complexity of spline functions, the speed and scalability of the original KAN were not satisfactory. Consequently, subsequent research explored the use of simpler basis functions to replace splines, thereby achieving higher efficiency. ChebyshevKAN (SS, 2024) incorporates Chebyshev polynomials to parametrize the learnable functions. FastKAN (Li, 2024) uses faster Gaussian radial basis functions to approximate third-order B-spline functions.

Moreover, KAN has been applied as alternatives to MLP in various domains. Convolutional KAN (Bodner et al., 2024) replaces the linear weight matrices in traditional convolutional networks with learnable spline function matrices. U-KAN (Li et al., 2024) integrates KAN layers into the U-Net architecture, demonstrating impressive accuracy and efficiency in several medical image segmentation tasks. KAN has also been used to bridge the gap between AI and science. Works such as PIKAN (Shukla et al., 2024) and PINN (Wang et al., 2024b) utilize KAN to build physics-informed machine learning models. This paper aims to introduce KAN into TSF and demonstrate the strong potential of KAN in representing time series data.

## 2.2 TIME SERIES FORECASTING

Traditional time series forecasting (TSF) methods, such as ARIMA (Zhang, 2003), can provide sufficient interpretability for the forecasting results but often fail to achieve satisfactory accuracy. In recent years, deep learning methods have dominated the field of TSF, mainly including CNN-based, Transformer-based, and MLP-based approaches. CNN-based models primarily apply convolution operations along the temporal dimension to extract temporal patterns. For example, MICN (Wang et al., 2023) and TimesNet (Wu et al., 2023) enhance the precision of sequence modeling by adjusting the receptive field to capture both short-term and long-term views within the sequences. ModernTCN (donghao & wang xue, 2024) advocates using large convolution kernels along the temporal dimension and capture both cross-time and cross-variable dependencies. Compared to CNN-based methods, which have limited receptive field, Transformer-based methods offer global modeling capabilities, making them more suitable for handling long and complex sequence data. They have become the cornerstone of modern time series forecasting. Informer (Zhou et al., 2021) is one of the early implementations of Transformer models in TSF, making efficient forecasting possible by carefully modifying the internal Transformer architecture. PatchTST (Nie et al., 2023) divides the sequence into multiple patches along the temporal dimension, which are then fed into the Transformer, establishing it as an important benchmark in the time series domain. In contrast, iTransformer (Liu et al., 2024b) treats each variable as an independent token to capture cross-variable dependencies in multivariate time series. However, Transformer-based methods face challenges due to the large number of parameters and high memory consumption. Recent research on MLP-based methods has shown that with appropriately designed architectures leveraging prior knowledge, simple MLPs can outperform complex Transformer-based methods. DLinear (Zeng et al., 2023), for instance, preprocesses sequences using a trend-season decomposition strategy. FITS (Xu et al., 2024b) performs linear transformations in the frequency domain, while TimeMixer (Wang et al., 2024a) uses MLP to facilitate information interaction at different scales. These MLP-based methods have demonstrated strong performance regarding both forecasting accuracy and efficiency. Unlike the aforementioned methods, this paper introduces the novel KAN to TSF to represent time series data more accurately. It also proposes a well-designed Decomposition-Learning-Mixing architecture to fully unlock the potential of KAN for time series forecasting.

## 2.3 TIME SERIES DECOMPOSITION

Real-world time series often consist of various underlying patterns. To leverage the characteristics of different patterns, recent approaches tend to decompose the series into multiple subcomponents, including trend-seasonal decomposition, multi-scale decomposition, and multi-period decomposition. DLinear (Zeng et al., 2023) employs moving averages to decouple the seasonal and trend components. SCINet (Liu et al., 2022) uses a hierarchical downsampling tree to iteratively extract and exchange information at multiple temporal resolutions. TimeMixer (Wang et al., 2024a) follows a fine-to-coarse principle to decompose the sequence into multiple scales across different

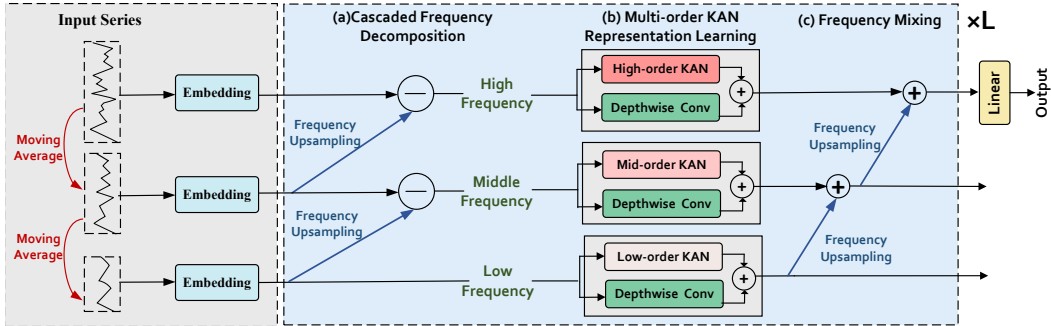

Figure 1: The architecture of TimeKAN, which mainly consists of Cascaded Frequency Decomposition block, Multi-order KAN Representation Learning block, and Frequency Mixing block. Here, we divide the frequency range of the time series into three frequency bands as an example.

time spans and further splits each scale into seasonal and periodic components. TimesNet (Wu et al., 2023) and PDF (Dai et al., 2024) utilize Fourier periodic analysis to decouple sequence into multiple sub-period sequences based on the calculated period. Inspired by these works, this paper proposes a novel Decomposition-Learning-Mixing architecture, which examines time series from a multi-frequency perspective to accurately model the complex patterns within time series.

## 3 TIMEKAN

### 3.1 OVERALL ARCHITECTURE

Given a historical multivariate time series input $\mathbf{X} \in \mathbb{R}^{N \times T}$, the aim of time series forecasting is to predict the future output series $\mathbf{X}_O \in \mathbb{R}^{N \times F}$, where $T, F$ is the look-back window length and the future window length, and $N$ represents the number of variates. In this paper, we propose TimeKAN to tackle the challenges arising from the complex mixture of multi-frequency components in time series. The overall architecture of TimeKAN is shown in Figure 1. We adopt variate-independent manner (Nie et al., 2023) to predict each univariate series independently. Each univariate input time series is denoted as $X \in \mathbb{R}^T$ and we consider univariate time series as the instance in the following calculation. In our TimeKAN, the first step is to progressively remove the relatively high-frequency components using moving averages and generate multi-level sequences followed by projecting each sequence into a high-dimensional space. Next, adhering to the Decomposition-Learning-Mixing architecture design principle, we first design Cascaded Frequency Decomposition (CFD) blocks to obtain sequence representations for each frequency band, adopting a bottom-up cascading approach. Then, we propose Multi-order KAN Representation Learning (M-KAN) blocks to learn and represent specific temporal patterns within each frequency band. Finally, Frequency Mixing blocks recombine the frequency bands into the original format, ensuring that the Decomposition-Learning-Mixing process is repeatable. More details about our TimeKAN are described as follow.

### 3.2 HIERARCHICAL SEQUENCE PREPROCESSING

Assume that we divide the frequency range of raw time series $X$ into predefined $k$ frequency bands. We first use moving average to progressively remove the relatively high-frequency components and generate multi-level sequences $\{x_1, \cdots, x_k\}$, where $x_i \in \mathbb{R}^{\frac{T}{d^{i-1}}} (i \in \{1, \cdots, k\})$. $x_1$ is equal to the input series $X$ and $d$ denotes the length of moving average window. The process of producing multi-level sequences is as follows:

$$x_i = \text{AvgPool}(\text{Padding}(x_{i-1})) \tag{1}$$

After obtaining the multi-level sequences, each sequence is independently embedded into a higher dimension through a Linear layer:

$$x_i = \text{Linear}(x_i) \tag{2}$$

where $x_i \in \mathbb{R}^{\frac{T}{a^{i-1}} \times D}$ and $D$ is embedding dimension. We define $x_1$ as the highest level sequence and $x_k$ as the lowest level sequence. Notably, each lower-level sequence is derived from the sequence one level higher by removing a portion of the high-frequency information. The above process is a preprocessing process and only occurs once in TimeKAN.

## 3.3 CASCADED FREQUENCY DECOMPOSITION

Real-world time series are often composed of multiple frequency components, with the low-frequency component representing long-term changes in the time series and the high-frequency component representing short-term fluctuations or unexpected events. These different frequency components complement each other and provide a comprehensive perspective for accurately modeling time series. Therefore, we design the Cascaded Frequency Decomposition (CFD) block to accurately decompose each frequency component in a cascade way, thus laying the foundation for accurately modeling different frequency components.

The aim of CFD block is to obtain the representation of each frequency component. Here, we take obtaining the representation of the $i$-th frequency band as an example. To achieve it, we first employ the Fast Fourier Transform (FFT) to obtain the representation of $x_{i+1}$ in the frequency domain. Then, Zero-Padding is used to extend the length of the frequency domain sequence, so that it can have the same length as the upper sequence $x_i$ after transforming back to the time domain. Next, we use Inverse Fast Fourier Transform (IFFT) to transform it back into the time domain. We refer to this upsampling process as Frequency Upsampling, which ensures that the frequency information remains unchanged before and after the upsampling. The process of Frequency Upsampling can be described as:

$$\hat{x}_i = \text{IFFT}(\text{Padding}(\text{FFT}(x_{i+1}))) \tag{3}$$

Here, $\hat{x}_i$ and $x_i$ have the same sequence length. Notably, compared to $x_i$, $\hat{x}_i$ lacks the $i$-th frequency component. The reason is that $x_{i+1}$ is originally formed by removing $i$-th frequency component from $x_i$ in the hierarchical sequence preprocessing and $x_{i+1}$ is now transformed into $\hat{x}_i$ through a lossless frequency conversion process, thereby aligning length with $x_i$ in the time domain. Therefore, to get the series representation of the $i$-th frequency component $f_i$ in time domain, we only need to get the residuals between $x_i$ and $\hat{x}_i$:

$$f_i = x_i - \hat{x}_i \tag{4}$$

## 3.4 MULTI-ORDER KAN REPRESENTATION LEARNING

Given the multi-level frequency component representation $\{f_1, \cdots, f_k\}$ generated by the CFD block, we propose Multi-order KAN Representation Learning (M-KAN) blocks to learn specific representations and temporal dependencies at each frequency. M-KAN adopts a dual-branch parallel architecture to separately model temporal representation learning and temporal dependency learning in a frequency-specific way, using Multi-order KANs to learn the representation of each frequency component and employing Depthwise Convolution to capture the temporal dependency. The details of Depthwise Convolution and Multi-order KAN will be given as follows.

**Depthwise Convolution**  To separate the modeling of temporal dependency from learning sequence representation, we adopt a specific type of group convolution known as Depthwise Convolution, in which the number of groups matches the embedding dimension. Depthwise Convolution employs $D$ groups of convolution kernels to perform independent convolution operations on the series of each channel. This allows the model to focus on capturing temporal patterns without interference from inter channel relationships. The process of Depthwise Convolution is:

$$f_{i,1} = Conv_{D \to D}(f_i, \text{group} = D) \tag{5}$$

**Multi-order KANs**  Compared with traditional MLP, KAN replaces linear weights with learnable univariate functions, allowing complex nonlinear relationships to be modeled with fewer parameters and greater interpretability. (Xu et al., 2024a). Assume that KAN is composed of $L + 1$ layer neurons and the number of neurons in layer $l$ is $n_l$. The transmission relationship between the $j$-th neuron in layer $l + 1$ and all neurons in layer $l$ can be expressed as $z_{l+1,j} = \sum_{i=1}^{n_l} \phi_{l,j,i}(z_{l,i})$, where $z_{l+1,j}$ is the $j$-th neuron at layer $l + 1$ and $z_{l,i}$ is the $i$-th neuron at layer $l$. We can simply understand

that each neuron is connected to other neurons in the previous layer through a learnable univariate function $\phi$. The vanilla KAN (Liu et al., 2024c) employs spline function as the learnable univariate basic functions $\phi$, but suffering from the complex recursive computation process, which hinders the efficiency of KAN. Here, we adopt ChebyshevKAN (SS, 2024) to learn the representation of each frequency component, i.e., channel learning. ChebyshevKAN is constructed from linear combinations of Chebyshev polynomial. That is, using the linear combination of Chebyshev polynomial with different order to generate learnable univariate function $\phi$. The Chebyshev polynomial is defined by:

$$T_n(x) = \cos(n \arccos(x)) \tag{6}$$

where $n$ is the highest order of Chebyshev polynomials and the complexity of Chebyshev polynomials is increasing with increasing order. A 1-layer ChebyshevKAN applied to channel dimension can be expressed as:

$$\phi_o(x) = \sum_{j=1}^{D} \sum_{i=0}^{n} \Theta_{o,j,i} T_i(\tanh(x_j)) \tag{7}$$

$$\text{KAN}(x) = \left\{ \begin{array}{c} \phi_1(x) \\ \cdots \\ \phi_D(x) \end{array} \right\} \tag{8}$$

where $o$ is the index of output neuron and $\Theta \in \mathbb{R}^{D \times D \times (n+1)}$ are the learnable coefficients used to linearly combine the Chebyshev polynomials. It is worth noting that the frequency components within the time series exhibit increasingly complex temporal dynamics as the frequency increases, necessitating a network with stronger representation capabilities to learn these characteristics. ChebyshevKAN allows for the adjustment of the highest order of Chebyshev polynomials $n$ to enhance its representation ability. Therefore, from the low-frequency to high-frequency components, we adopt an increasing order of Chebyshev polynomials to align the frequency components with the complexity of the KAN, thereby accurately learning the representations of different frequency components. We refer to this group of KANs with varying highest Chebyshev polynomials orders as Multi-order KANs. We set an lower bound order $b$, and the representation learning process for $x_i$ can be expressed as:

$$f_{i,2} = \text{KAN}(f_i, \text{order} = b + k - i) \tag{9}$$

The final output of the M-KAN block is the sum of the outputs from the Multi-order KANs and the Depthwise Convolution.

$$\hat{f}_i = f_{i,1} + f_{i,2} \tag{10}$$

## 3.5 Frequency Mixing

After specifically learning the representation of each frequency component, we need to re-transform the frequency representations into the form of multi-level sequences before entering next CFD block, ensuring that the Decomposition-Learning-Mixing process is repeatable. Therefore, we designed Frequency Mixing blocks to convert the frequency component at $i$-th level $\hat{f}_i$ into multi-level sequences $x_i$, enabling it to serve as input for the next CFD block. To transform the frequency component at $i$-th level $\hat{f}_i$ into multi-level sequences $x_i$, we simply need to to supplement the frequency information from levels $i + 1$ to $k$ back into the $i$-th level. Thus, we employ Frequency Upsampling again to incrementally reintegrate the information into the higher frequency components:

$$x_i = \text{IFFT}(\text{Padding}(\text{FFT}(x_{i+1}))) + f_i \tag{11}$$

For the last Frequency Mixing block, we extract the highest-level sequence $x_1$ and use a simple linear layer to produce the forecasting results $X_O$.

$$X_O = \text{Linear}(x_1) \tag{12}$$

Due to the use of a variate-independent strategy, we also need to stack the predicted results of all variables together to obtain the final multivariate prediction $\mathbf{X_O}$.

Table 1: Full results of the multivariate long-term forecasting result comparison. The input sequence length is set to 96 for all baselines and the prediction lengths $F \in \{96, 192, 336, 720\}$. *Avg* means the average results from all four prediction lengths.

| Models | | TimeKAN Ours | | TimeMixer 2024a | | iTransformer 2024b | | Time-FFM 2024a | | PatchTST 2023 | | TimesNet 2023 | | MICN 2023 | | DLinear 2023 | | FreTS 2024 | | FiLM 2022a | | FEDformer 2022b | | Autoformer 2021 | |
|---|---|---|---|---|---|---|---|---|---|---|---|---|---|---|---|---|---|---|---|---|---|---|---|---|---|
| Metric | | MSE | MAE | MSE | MAE | MSE | MAE | MSE | MAE | MSE | MAE | MSE | MAE | MSE | MAE | MSE | MAE | MSE | MAE | MSE | MAE | MSE | MAE | MSE | MAE |
| ETTh1 | 96 | **0.367** | **0.395** | 0.385 | 0.402 | 0.386 | 0.405 | 0.385 | 0.400 | 0.460 | 0.447 | 0.384 | 0.402 | 0.426 | 0.446 | 0.397 | 0.412 | 0.395 | 0.407 | 0.438 | 0.433 | 0.395 | 0.424 | 0.449 | 0.459 |
| | 192 | **0.414** | **0.420** | 0.443 | 0.430 | 0.441 | 0.436 | 0.439 | 0.430 | 0.512 | 0.477 | 0.436 | 0.429 | 0.454 | 0.464 | 0.446 | 0.441 | 0.490 | 0.477 | 0.494 | 0.466 | 0.469 | 0.470 | 0.500 | 0.482 |
| | 336 | **0.445** | **0.434** | 0.512 | 0.470 | 0.487 | 0.458 | 0.480 | 0.449 | 0.546 | 0.496 | 0.638 | 0.469 | 0.493 | 0.487 | 0.489 | 0.467 | 0.510 | 0.480 | 0.547 | 0.495 | 0.490 | 0.477 | 0.521 | 0.496 |
| | 720 | **0.444** | **0.459** | 0.497 | 0.476 | 0.503 | 0.491 | 0.462 | 0.456 | 0.544 | 0.517 | 0.521 | 0.500 | 0.526 | 0.526 | 0.513 | 0.510 | 0.568 | 0.538 | 0.586 | 0.538 | 0.598 | 0.544 | 0.514 | 0.512 |
| | Avg | **0.417** | **0.427** | 0.459 | 0.444 | 0.454 | 0.447 | 0.442 | 0.434 | 0.516 | 0.484 | 0.495 | 0.450 | 0.475 | 0.480 | 0.461 | 0.457 | 0.491 | 0.475 | 0.516 | 0.483 | 0.498 | 0.484 | 0.496 | 0.487 |
| ETTh2 | 96 | 0.290 | 0.340 | **0.289** | 0.342 | 0.297 | 0.349 | 0.301 | 0.351 | 0.308 | 0.355 | 0.340 | 0.374 | 0.372 | 0.424 | 0.340 | 0.394 | 0.332 | 0.387 | 0.322 | 0.364 | 0.358 | 0.397 | 0.346 | 0.388 |
| | 192 | **0.375** | **0.392** | 0.378 | 0.397 | 0.380 | 0.400 | 0.378 | 0.397 | 0.393 | 0.405 | 0.402 | 0.414 | 0.492 | 0.492 | 0.482 | 0.479 | 0.451 | 0.457 | 0.405 | 0.414 | 0.429 | 0.439 | 0.456 | 0.452 |
| | 336 | 0.423 | 0.435 | 0.432 | 0.434 | 0.428 | 0.432 | **0.422** | **0.431** | 0.427 | 0.436 | 0.452 | 0.452 | 0.607 | 0.555 | 0.591 | 0.541 | 0.466 | 0.473 | 0.435 | 0.445 | 0.496 | 0.487 | 0.482 | 0.486 |
| | 720 | 0.443 | 0.449 | 0.464 | 0.464 | 0.427 | 0.445 | **0.427** | **0.444** | 0.436 | 0.450 | 0.462 | 0.468 | 0.824 | 0.655 | 0.839 | 0.661 | 0.485 | 0.471 | 0.445 | 0.457 | 0.463 | 0.474 | 0.515 | 0.511 |
| | Avg | 0.383 | **0.404** | 0.390 | 0.409 | 0.383 | 0.407 | **0.382** | 0.406 | 0.391 | 0.411 | 0.414 | 0.427 | 0.574 | 0.531 | 0.563 | 0.519 | 0.433 | 0.446 | 0.402 | 0.420 | 0.437 | 0.449 | 0.450 | 0.459 |
| ETTm1 | 96 | 0.322 | 0.361 | **0.317** | **0.356** | 0.334 | 0.368 | 0.336 | 0.369 | 0.352 | 0.374 | 0.338 | 0.375 | 0.365 | 0.387 | 0.346 | 0.374 | 0.337 | 0.374 | 0.353 | 0.370 | 0.379 | 0.419 | 0.505 | 0.475 |
| | 192 | **0.357** | **0.383** | 0.367 | 0.384 | 0.377 | 0.391 | 0.378 | 0.389 | 0.390 | 0.393 | 0.374 | 0.387 | 0.403 | 0.408 | 0.382 | 0.391 | 0.382 | 0.398 | 0.389 | 0.387 | 0.426 | 0.441 | 0.553 | 0.496 |
| | 336 | **0.382** | **0.401** | 0.391 | 0.406 | 0.426 | 0.420 | 0.411 | 0.410 | 0.421 | 0.414 | 0.410 | 0.411 | 0.436 | 0.431 | 0.415 | 0.415 | 0.420 | 0.423 | 0.421 | 0.408 | 0.445 | 0.459 | 0.621 | 0.537 |
| | 720 | **0.445** | **0.435** | 0.454 | 0.441 | 0.491 | 0.459 | 0.469 | 0.441 | 0.462 | 0.449 | 0.478 | 0.450 | 0.489 | 0.462 | 0.473 | 0.451 | 0.490 | 0.471 | 0.481 | 0.441 | 0.543 | 0.490 | 0.671 | 0.561 |
| | Avg | **0.376** | **0.395** | 0.382 | 0.397 | 0.407 | 0.410 | 0.399 | 0.402 | 0.406 | 0.407 | 0.400 | 0.406 | 0.423 | 0.422 | 0.404 | 0.408 | 0.407 | 0.417 | 0.412 | 0.402 | 0.448 | 0.452 | 0.588 | 0.517 |
| ETTm2 | 96 | **0.174** | **0.255** | 0.175 | 0.257 | 0.181 | 0.267 | 0.183 | 0.270 | 0.183 | 0.270 | 0.187 | 0.267 | 0.197 | 0.296 | 0.193 | 0.293 | 0.186 | 0.275 | 0.183 | 0.266 | 0.203 | 0.287 | 0.255 | 0.339 |
| | 192 | **0.239** | **0.299** | 0.240 | 0.302 | 0.250 | 0.309 | 0.247 | 0.308 | 0.255 | 0.314 | 0.249 | 0.309 | 0.284 | 0.361 | 0.284 | 0.361 | 0.259 | 0.323 | 0.248 | 0.305 | 0.269 | 0.328 | 0.281 | 0.340 |
| | 336 | **0.301** | **0.340** | 0.303 | 0.343 | 0.311 | 0.348 | 0.309 | 0.347 | 0.309 | 0.347 | 0.321 | 0.351 | 0.381 | 0.429 | 0.382 | 0.429 | 0.349 | 0.386 | 0.309 | 0.343 | 0.325 | 0.366 | 0.339 | 0.372 |
| | 720 | **0.395** | **0.396** | 0.392 | 0.396 | 0.412 | 0.407 | 0.406 | 0.404 | 0.412 | 0.404 | 0.408 | 0.403 | 0.549 | 0.522 | 0.558 | 0.525 | 0.559 | 0.511 | 0.410 | 0.400 | 0.421 | 0.415 | 0.433 | 0.432 |
| | Avg | **0.277** | **0.322** | 0.277 | 0.324 | 0.288 | 0.332 | 0.286 | 0.332 | 0.290 | 0.334 | 0.291 | 0.333 | 0.353 | 0.402 | 0.354 | 0.402 | 0.339 | 0.374 | 0.288 | 0.328 | 0.305 | 0.349 | 0.327 | 0.371 |
| Weather | 96 | **0.162** | **0.208** | 0.163 | 0.209 | 0.174 | 0.214 | 0.191 | 0.230 | 0.186 | 0.227 | 0.172 | 0.220 | 0.198 | 0.261 | 0.195 | 0.252 | 0.171 | 0.227 | 0.195 | 0.236 | 0.217 | 0.296 | 0.266 | 0.336 |
| | 192 | **0.207** | **0.249** | 0.211 | 0.254 | 0.236 | 0.267 | 0.234 | 0.265 | 0.219 | 0.261 | 0.219 | 0.261 | 0.239 | 0.299 | 0.237 | 0.295 | 0.218 | 0.280 | 0.239 | 0.271 | 0.276 | 0.336 | 0.307 | 0.367 |
| | 336 | **0.263** | **0.290** | 0.263 | 0.293 | 0.278 | 0.296 | 0.289 | 0.303 | 0.284 | 0.301 | **0.246** | 0.337 | 0.285 | 0.336 | 0.282 | 0.331 | 0.265 | 0.317 | 0.289 | 0.306 | 0.339 | 0.380 | 0.359 | 0.395 |
| | 720 | **0.338** | **0.340** | 0.344 | 0.348 | 0.358 | 0.347 | 0.362 | 0.350 | 0.356 | 0.349 | 0.365 | 0.359 | 0.351 | 0.388 | 0.345 | 0.382 | **0.326** | 0.351 | 0.351 | 0.360 | 0.403 | 0.428 | 0.419 | 0.428 |
| | Avg | **0.242** | **0.272** | 0.245 | 0.276 | 0.258 | 0.278 | 0.270 | 0.288 | 0.265 | 0.285 | 0.251 | 0.294 | 0.268 | 0.321 | 0.265 | 0.315 | 0.245 | 0.294 | 0.271 | 0.290 | 0.309 | 0.360 | 0.338 | 0.382 |
| Electricity | 96 | 0.174 | 0.266 | 0.153 | 0.245 | **0.148** | **0.240** | 0.198 | 0.282 | 0.190 | 0.296 | 0.168 | 0.272 | 0.180 | 0.293 | 0.210 | 0.302 | 0.171 | 0.260 | 0.198 | 0.274 | 0.193 | 0.308 | 0.201 | 0.317 |
| | 192 | 0.182 | 0.273 | 0.166 | 0.257 | **0.162** | **0.253** | 0.199 | 0.285 | 0.199 | 0.304 | 0.184 | 0.322 | 0.189 | 0.302 | 0.210 | 0.305 | 0.177 | 0.268 | 0.198 | 0.278 | 0.201 | 0.315 | 0.222 | 0.334 |
| | 336 | 0.197 | 0.286 | 0.185 | 0.275 | **0.178** | **0.269** | 0.212 | 0.298 | 0.217 | 0.319 | 0.198 | 0.300 | 0.198 | 0.312 | 0.223 | 0.319 | 0.190 | 0.284 | 0.217 | 0.300 | 0.214 | 0.329 | 0.231 | 0.443 |
| | 720 | 0.236 | 0.320 | 0.224 | **0.312** | 0.225 | 0.317 | 0.253 | 0.330 | 0.258 | 0.352 | 0.220 | 0.320 | 0.217 | 0.330 | 0.258 | 0.350 | 0.228 | 0.316 | 0.278 | 0.356 | 0.246 | 0.355 | 0.254 | 0.361 |
| | Avg | 0.197 | 0.286 | 0.182 | 0.272 | **0.178** | **0.270** | 0.270 | 0.288 | 0.216 | 0.318 | 0.193 | 0.304 | 0.196 | 0.309 | 0.225 | 0.319 | 0.192 | 0.282 | 0.223 | 0.302 | 0.214 | 0.327 | 0.227 | 0.338 |
| 1st Count | | **17** | **22** | 4 | 3 | 5 | 4 | 3 | 2 | 0 | 0 | 1 | 0 | 1 | 0 | 0 | 0 | 1 | 0 | 0 | 0 | 0 | 0 | 0 | 0 |

# 4 EXPERIMENTS

**Datasets** We conduct extensive experiments on six real-world time series datasets, including Weather, ETTh1, ETTh2, ETTm1, ETTm2 and Electricity for long-term forecasting. Following previous work (Wu et al., 2021), we split the ETT series dataset into training, validation, and test sets in a ratio of 6:2:2. For the remaining datasets, we adopt a split ratio of 7:1:2.

**Baseline** We carefully select eleven well-acknowledged methods in the field of long-term time series forecasting as our baselines, including (1) Transformer-based methods: Autoformer (2021), FEDformer (2022b), PatchTST (2023), iTransformer (2024b); (2) MLP-based methods: DLinear (2023) and TimeMixer (2024a) (3) CNN-based method: MICN (2023), TimesNet (2023); (4) Frequency-based methods: FreTS (2024) and FiLM (2022a). And a time series foundation model Time-FFM (2024a).

**Experimental Settings** To ensure fair comparisons, we adopt the same look-back window length $T = 96$ and the same prediction length $F = \{96, 192, 336, 720\}$. We utilize the L2 loss for model training and use Mean Square Error (MSE) and Mean Absolute Error (MAE) metrics to evaluate the performance of each method.

## 4.1 MAIN RESULTS

The comprehensive forecasting results are presented in Table 1, where the best results are highlighted in bold red and the second-best are underlined in blue. A lower MSE/MAE indicates a more accurate prediction result. We observe that TimeKAN demonstrates superior predictive performance across all datasets, except for the Electricity dataset, where iTransformer achieves the best result. This is due to iTransformer's use of channel-wise self-attention mechanisms to model inter-variable dependencies, which is particularly effective for high-dimensional datasets like Electricity. Additionally, both TimeKAN and TimeMixer perform consistently well in long-term forecasting tasks, showcasing the generalizability of well-designed time-series decomposition architectures for accurate predictions. Compared with other state-of-the-art methods, TimeKAN introduces a novel

Table 2: Ablation study of the Frequency Upsampling. The best results are in **bold**.

| Datasets | ETTh1 | | ETTh2 | | ETTm1 | | ETTm2 | | Weather | | Electricity | |
|---|---|---|---|---|---|---|---|---|---|---|---|---|
| Metric | MSE | MAE | MSE | MAE | MSE | MAE | MSE | MAE | MSE | MAE | MSE | MAE |
| Linear Mapping | 0.401 | 0.413 | 0.312 | 0.362 | 0.328 | 0.365 | 0.180 | 0.263 | 0.164 | 0.211 | 0.184 | 0.275 |
| Linear Interpolation | 0.383 | 0.398 | 0.296 | 0.347 | 0.336 | 0.370 | 0.181 | 0.263 | 0.165 | 0.210 | 0.196 | 0.277 |
| Transposed Convolution | 0.377 | 0.407 | **0.290** | 0.344 | 0.326 | 0.366 | 0.178 | 0.261 | 0.163 | 0.211 | 0.188 | 0.274 |
| **Frequency Upsamping** | **0.367** | **0.395** | **0.290** | **0.340** | **0.322** | **0.361** | **0.174** | **0.255** | **0.162** | **0.208** | **0.174** | **0.266** |

Table 3: Ablation study of the Multi-order KANs. The best results are in **bold**.

| Datasets | ETTh1 | | ETTh2 | | ETTm1 | | ETTm2 | | Weather | |
|---|---|---|---|---|---|---|---|---|---|---|
| Metric | MSE | MAE | MSE | MAE | MSE | MAE | MSE | MAE | MSE | MAE |
| MLPs | 0.376 | 0.397 | 0.298 | 0.348 | **0.319** | **0.361** | 0.178 | 0.264 | **0.162** | 0.211 |
| Fixed Low-order KANs | 0.376 | 0.398 | 0.292 | 0.341 | 0.327 | 0.366 | 0.175 | 0.257 | 0.164 | 0.211 |
| Fixed High-order KANs | 0.380 | 0.407 | 0.310 | 0.363 | 0.327 | 0.269 | 0.176 | 0.257 | 0.164 | 0.212 |
| **Multi-order KANs** | **0.367** | 0.395 | **0.290** | **0.340** | 0.322 | **0.361** | **0.174** | **0.255** | **0.162** | **0.208** |

Decomposition-Learning-Mixing framework, closely integrating the characteristics of Multi-order KANs with this hierarchical architecture, enabling superior performance in a wide range of long-term forecasting tasks.

## 4.2 ABLATION STUDY

In this section, we investigate several key components of TimeKAN, including Frequency Upsampling, Depthwise Convolution and Multi-order KANs.

**Frequency Upsampling** To investigate the effectiveness of Frequency Upsampling, we compared it with three alternative upsampling methods that may not preserve frequency information before and after transformation: (1) Linear Mapping; (2) Linear Interpolation; and (3) Transposed Convolution. As shown in Table 2, replacing Frequency Upsampling with any of these three methods resulted in a decline in performance. This indicates that these upsampling techniques fail to maintain the integrity of frequency information after transforming, leading to the Decomposition-Learning-Mixing framework ineffective. This strongly demonstrates that the chosen Frequency Upsampling, as a non-parametric method, is an irreplaceable component of the TimeKAN framework.

**Multi-order KANs** We designed the following modules to investigate the effectiveness of Multi-order KANs: (1) MLPs, which means using MLP to replace each KAN; (2) Fixed Low-order KANs, which means using a KAN of order 2 at each frequency level; and (3) Fixed High-order KANs, which means using a KAN of order 5 at each frequency level. The comparison results are shown in Table 3. Overall, Multi-order KANs achieved the best performance. Compared to MLPs, Multi-order KANs perform significantly better, demonstrating that well-designed KANs possess stronger representation capabilities than MLPs and are a compelling alternative. Both Low-order KANs and High-order KANs performed worse than Multi-order KANs, indicating the validity of our design choice to incrementally increase the order of KANs to adapt to the representation of different frequency components. Thus, the learnable functions of KANs are indeed a double-edged sword; achieving satisfactory results requires selecting the appropriate level of function complexity for specific tasks.

**Depthwise Convolution** To assess the effectiveness of Depthwise Convolution, we replace it with the following choice: (1) w/o Depthwise Convolution; (2) Standard Convolution; (3) Multi-head Self-Attention. The results are shown in Table 4. Overall, Depthwise Convolution is the best choice. We clearly observe that removing Depthwise Convolution or replacing it with Multi-head Self-Attention leads to a significant drop in performance, highlighting the effectiveness of using convolution to learn temporal dependencies. When Depthwise Convolution is replaced with Standard

Table 4: Ablation study of the Depthwise Convolution. The best results are in **bold**.

| Datasets | ETTh1 | | ETTh2 | | ETTm1 | | ETTm2 | | Weather | |
|---|---|---|---|---|---|---|---|---|---|---|
| Metric | MSE | MAE | MSE | MAE | MSE | MAE | MSE | MAE | MSE | MAE |
| w/o Depthwise Conv | 0.379 | 0.397 | 0.296 | 0.343 | 0.337 | 0.373 | 0.180 | 0.263 | 0.168 | 0.211 |
| Standard Conv | **0.364** | **0.393** | 0.295 | 0.345 | 0.323 | 0.364 | 0.180 | 0.264 | **0.162** | 0.210 |
| Self-Attention | 0.377 | 0.406 | 0.293 | 0.342 | 0.329 | 0.365 | 0.184 | 0.272 | 0.174 | 0.225 |
| **Depthwise Conv** | 0.367 | 0.395 | **0.290** | **0.340** | **0.322** | **0.361** | **0.174** | **0.255** | 0.162 | **0.208** |

Figure 2: Comparison of forecasting performance between TimeKAN and other three models with varying look-back windows on ETTm2 and Weather datasets. The look-back windows are selected to be $T \in \{48, 96, 192, 336, 512, 720\}$, and the prediction length is fixed to $F = 96$.

Convolution, there are declines in most metrics, which implies that focusing on extracting temporal dependencies individually with Depthwise Convolution, without interference from inter-channel relationships, is a reasonable design.

**Varing Look-back Window** In principle, extending the look-back window can provide more information for predicting future, leading to a potential improvement in forecasting performance. A effective long-term TSF method equipped with a strong temporal relation extraction capability should be able to improve forecasting performance when look-back window length increasing (Zeng et al., 2023). As a model based on frequency decomposition learning, TimeKAN should achieve better predictive performance as the look-back window lengthens, since more incremental frequency information is available for prediction. To demonstrate that TimeKAN benefits from a larger look-back window, we select look-back window lengths from $T = \{48, 96, 192, 336, 512, 720\}$ while keeping the prediction length fixed at 96. As demonstrated in Figure 2, our TimeKAN consistently reduces the MSE scores as the look-back window increases, indicating that TimeKAN can effectively learn from long time series.

## 4.3 MODEL EFFICIENCY

We compare TimeKAN with MLP-based method TimeMier and Transformer-based methods iTransformer and PatchTST, in terms of model parameters and Multiply-Accumulate Operations (MACs), to validate that TimeKAN is a lightweight and efficient architecture. To ensure a fair comparison, we fix the prediction length $F = 96$ and input length $T = 96$, and set the input batch size to 32. The comparison results are summarized in Table 5. It is clear that our TimeKAN demonstrates significant advantages in both model parameter size and MACs, particularly when compared to Transformer-based models. For instance, on the Electricity dataset, the parameter count of PatchTST is nearly 295 times that of TimeKAN, and its MACs are almost 118 times greater. Even when compared to the relatively lightweight MLP-based method TimeMixer, TimeKAN shows superior efficiency. On the Weather dataset, TimeKAN requires only 20.05% of the parameters needed by TimeMixer and only 36.14% of the MACs. This remarkable efficiency advantage is primarily attributed to the lightweight architectural design. The main computations of the TimeKAN model are concentrated

Table 5: A comparison of model parameters (Params) and multiply-accumulate operations (MACs) for TimeKAN and three other models. To ensure a fair comparison, we fix the prediction length $F = 96$ and the input length $T = 96$, and set the input batch size to 32. The lowest computational cost is highlighted in **bold**.

| Datasets | ETTH1 | | ETTH2 | | ETTm1 | | ETTm2 | | Weather | | Electricity | |
|---|---|---|---|---|---|---|---|---|---|---|---|---|
| Metric | Params | MACs | Params | MACs | Params | MACs | Params | MACs | Params | MACs | Params | MACs |
| TimeMixer | 75.50K | 20.37M | 75.50K | 20.37M | 75.50K | 20.37M | 77.77K | 24.18M | 104.43K | 82.62M | 106.83K | 1.26G |
| iTransformer | 841.57K | 77.46M | 224.22K | 19.86M | 224.22K | 19.86M | 224.22K | 19.86M | 4.83M | 1.16G | 4.83M | 16.29G |
| PatchTST | 3.75M | 5.90G | 10.06M | 17.66G | 3.75M | 5.90G | 10.06M | 17.66G | 6.90M | 35.30G | 6.90M | 539.38G |
| **TimeKAN** | **12.84K** | **7.63M** | **15.00K** | **8.02M** | **14.38K** | **7.63M** | **38.12K** | **16.66M** | **20.94K** | **29.86M** | **23.34K** | **456.50M** |

in the M-KAN block, and the Depthwise Convolution we employed significantly reduces the number of parameters through grouped operations. Additionally, the powerful representation capabilities afforded by Multi-order KANs allow us to represent time series with very few neurons. Therefore, we cannot overlook that TimeKAN achieves outstanding forecasting performance while requiring minimal computational resources.

## 5 CONCLUSION

We proposed an efficient KAN-based Frequency Decomposition Learning architecture (TimeKAN) for long-term time series forecasting. Based on Decomposition-Learning-Mixing architecture, TimeKAN obtains series representations for each frequency band using a Cascaded Frequency Decomposition blocks. Additionally. a Multi-order KAN Representation Learning blocks further leverage the high flexibility of KAN to learn and represent specific temporal patterns within each frequency band. Finally, Frequency Mixing blocks recombine the frequency bands into the original format. Extensive experiments on real-world datasets demonstrate that TimeKAN achieves the state of the art forecasting performance and extremely lightweight computational consumption.

## ACKNOWLEDGEMENTS

This work is supported by Shanghai Artificial Intelligence Laboratory. This work was done during Songtao Huang's internship at Shanghai Artificial Intelligence Laboratory.

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

# A   ADDITIONAL MODEL ANALYSIS

Table 6: Full comparison results of model parameters (Params) and multiply-accumulate operations (MACs) for TimeKAN and other models. To ensure a fair comparison, we fix the prediction length $F = 96$ and the input length $T = 96$, and set the input batch size to 32. The lowest computational cost is highlighted in **bold**.

| Datasets | ETTH1 | | ETTH2 | | ETTm1 | | ETTm2 | | Weather | | Electricity | |
|---|---|---|---|---|---|---|---|---|---|---|---|---|
| Metric | Params | MACs | Params | MACs | Params | MACs | Params | MACs | Params | MACs | Params | MACs |
| TimeMixer | 75.50K | 20.37M | 75.50K | 20.37M | 75.50K | 20.37M | 77.77K | 24.18M | 104.43K | 82.62M | 106.83K | 1.26G |
| iTransformer | 841.57K | 77.46M | 224.22K | 19.86M | 224.22K | 19.86M | 224.22K | 19.86M | 4.83M | 1.16G | 4.83M | 16.29G |
| PatchTST | 3.75M | 5.90G | 10.06M | 17.66G | 3.75M | 5.90G | 10.06M | 17.66G | 6.90M | 35.30G | 6.90M | 539.38G |
| TimesNet | 605.48K | 18.13G | 1.19M | 36.28G | 4.71M | 144G | 1.19M | 36.28G | 1.19M | 36.28G | 150.30M | 4.61T |
| MICN | 25.20M | 71.95G | 25.20M | 71.95G | 25.20M | 71.95G | 25.20M | 71.95G | 111.03K | 295.07M | 6.64M | 19.5G |
| Dlinear | 18.62K | 0.6M | 18.62K | 0.6M | 18.62K | 0.6M | 18.62K | 0.6M | 18.62K | 0.6M | 18.62K | 0.6M |
| FreTS | 3.24M | 101.46M | 3.24M | 101.46M | 3.24M | 101.46M | 3.24M | 101.46M | 3.24M | 101.46M | 3.24M | 101.46M |
| FILM | 12.58M | 2.82G | 12.58M | 2.82G | 12.58M | 2.82G | 12.58M | 2.82G | 12.58M | 8.46G | 12.58M | 8.46G |
| FEDFormer | 23.38M | 24.96G | 23.38M | 24.96G | 23.38M | 24.96G | 23.38M | 24.96G | 23.45M | 25.23G | 24.99M | 30.89G |
| AutoFormer | 10.54M | 22.82G | 10.54M | 22.82G | 10.54M | 22.82G | 10.54M | 22.82G | 10.61M | 23.08G | 12.14M | 28.75G |
| TimeKAN | 12.84K | 7.63M | 15.00K | 8.02M | 14.38K | 7.63M | 38.12K | 16.66M | 20.94K | 29.86M | 23.34K | 456.50M |

## A.1   COMPUTATIONAL COMPLEXITY ANALYSIS

In our TimeKAN, the main computational complexity lies in Fast Fourier Transform (FFT), Depthwise Convolution block and Multi-order KAN block. Consider a time series with length $L$ and the hidden state of each time point is $D$. For FFT, the computation complexity is $\mathcal{O}(L \, logL)$. For Depthwise Convolution block, if we set the convolutional kernel to $M$ and stride to 1, the complexity is $\mathcal{O}(LDM)$. Finally, assuming that the highest order of Chebyshev polynomials is $K$, the complexity of Multi-order KAN block is $\mathcal{O}(LD^2K)$. Since $M, D, K$ are constants that are independent of the input length $L$, the computational complexity of both the Depthwise Convolution block and the Multi-order KAN block can be reduced to $\mathcal{O}(L)$, which is linear about the sequence length. In summary, the overall computational complexity is $\max(\mathcal{O}(L \, logL), \mathcal{O}(L) = \mathcal{O}(L \, logL)$. When the input is a multivariate sequence with $M$ variables, the computational complexity will expand to $\mathcal{O}(ML \, logL)$ due to our variable-independent strategy.

## A.2   MODEL EFFICIENCY

Here, we provide the complete results of model efficiency in terms of parameters and MACs in Table 6 . As can be seen, except for DLinear, our TimeKAN consistently demonstrates a significant advantage in both parameter count and MACs compared to any other model. DLinear is a model consisting of only a single linear layer, which makes it the most lightweight in terms of parameters and MACs. However, the performance of DLinear already shows a significant gap when compared to state-of-the-art methods. Therefore, our TimeKAN actually achieves superior performance in both forecasting accuracy and efficiency.

## A.3   ERROR BARS

To evaluate the robustness of TimeKAN, we repeated the experiments on three randomly selected seeds and compared it with the second-best model (TimeMixer). We report the mean and standard deviation of the results across the three experiments, as well as the confidence level of TimeKAN's superiority over TimeMixer. The results are averaged over four prediction horizons (96, 192, 336, and 720). As shown in the Table 7, in most cases, we have over 90% confidence that TimeKAN outperforms the second-best model and demonstrates good robustne of TimeKAN.

Table 7: Standard deviation and statistical tests for our TimeKAN method and second-best method (TimeMixer) on five datasets.

| Metric | MSE | | | MAE | | |
|---|---|---|---|---|---|---|
| Dataset | TimeKAN | TimeMixer | Confidence | TimeKAN | TimeMixer | Confidence |
| ETTh1 | **0.422±0.004** | 0.462±0.006 | 99% | **0.430±0.002** | 0.448±0.004 | 99% |
| ETTh2 | **0.387±0.003** | 0.392±0.003 | 99% | **0.408±0.003** | 0.412±0.004 | 90% |
| ETTm1 | **0.378±0.002** | 0.386±0.003 | 99% | **0.396±0.001** | 0.399±0.001 | 99% |
| ETTm2 | 0.278±0.001 | 0.278±0.001 | — | **0.324±0.001** | 0.325±0.001 | 90% |
| Weather | **0.243±0.001** | 0.245±0.001 | 99% | **0.273±0.001** | 0.276±0.001 | 99% |

Table 8: Comparison on the Electricity dataset when the look back window is expanded to 512.

| Models | 96 | | 192 | | 336 | | 720 | |
|---|---|---|---|---|---|---|---|---|
| | MSE | MAE | MSE | MAE | MSE | MAE | MSE | MAE |
| MOMENT | 0.136 | 0.233 | 0.152 | 0.247 | 0.167 | 0.264 | 0.205 | 0.295 |
| TimeMixer | 0.135 | 0.231 | **0.149** | **0.245** | 0.172 | 0.268 | **0.203** | 0.295 |
| TimeKAN | **0.133** | **0.230** | **0.149** | 0.247 | **0.165** | **0.261** | **0.203** | **0.294** |

## A.4 FREQUENCY LEARNING WITH LONGER WINDOW

In Table 1, TimeKAN performs relatively poorly on the Electricity dataset. We infer that its poor performance on the electricity dataset is due to the overly short look-back window ($T = 96$), which cannot provide sufficient frequency information. To verify this, we compare the average number of effective frequency components under a specific look-back window. Specifically, we randomly select a sequence of length $T$ from the electricity dataset and transform it into the frequency domain using FFT. We define effective frequencies as those with amplitudes greater than 0.1 times the maximum amplitude. Then, we take the average number of effective frequencies obtained across all variables to reflect the amount of effective frequency information provided by the sequence. When $T = 96$ (the setting in this paper), the average number of effective frequencies is 10.69. When we extend the sequence length to 512, the average number of effective frequencies becomes 19.74. Therefore, the effective frequency information provided by 512 time steps is nearly twice that of 96 time steps. This indicates that $T = 96$ loses a substantial amount of effective information.

To validate whether using $T = 512$ allows us to leverage more frequency information, we extend the look-back window of TimeKAN to 512 on the electricity dataset and compare it with the state-of-the-art methods TimeMixer and time series foundatiom model MOMENT (Goswami et al., 2024). The results are shown in Table 8. Although TimeKAN performs significantly worse than TimeMixer when $T = 96$, it achieves the best performance on the electricity dataset when the look-back window is extended to 512. This also demonstrates that TimeKAN can benefit significantly from richer frequency information.

## A.5 IMPACT OF NUMBER OF FREQUENCY BANDS

To explore the impact of the number of frequency bands on performance, we set the number of frequency bands to 2, 3, 4, and 5. The effects of different frequency band divisions on performance are shown in the Table 9. As we can see, in most cases, dividing the frequency bands into 3 or 4 layers yields the best performance. This aligns with our prior intuition: dividing into two bands results in excessive frequency overlap, while dividing into five bands leads to too little information within each band, making it difficult to accurately model the information within that frequency range.

Table 9: Impact of number of frequency bands on performance under the 96-to-96 prediction setting.

| Number of Frequency | ETTh2 | | Weather | | Electricity | |
|---|---|---|---|---|---|---|
| | MSE | MAE | MSE | MAE | MSE | MAE |
| 2 | 0.292 | 0.340 | 0.164 | 0.209 | 0.183 | 0.270 |
| 3 | **0.290** | **0.339** | 0.163 | 0.209 | 0.177 | 0.268 |
| 4 | **0.290** | 0.340 | **0.162** | **0.208** | **0.174** | **0.266** |
| 5 | 0.295 | 0.346 | 0.164 | 0.211 | 0.177 | 0.273 |

# B  MATHEMATICAL DETAILS

## B.1  KOLMOGOROV-ARNOLD NETWORK

Kolmogorov-Arnold representation theorem states that any multivariate continuous function can be expressed as a combination of univariate functions and addition operations. More specifically, a multivariate continuous function $g : [0,1]^n \Rightarrow \mathbb{R}$ can be defined as:

$$g(x) = g(x_1, \cdots, x_n) = \sum_{i=1}^{2n+1} \Phi_i \Big( \sum_{j=1}^{n} \phi_{ij}(x_j) \Big) \tag{13}$$

where $\phi_{ij}$ and $\Phi_i$ are univariate functions. Following the pattern of MLP, Kolmogorov-Arnold Network (KAN) (Liu et al., 2024c) extends the Kolmogorov-Arnoldtheorem to deep representations, i.e., stacked multilayer Kolmogorov-Arnold representations. Assume that KAN is composed of $L+1$ layer neurons and the number of neurons in layer $l$ is $n_l$. The transmission relationship between the $j$-th neuron in layer $l+1$ and all neurons in layer $l$ can be expressed as:

$$x_{l+1,j} = \sum_{i=1}^{n_l} \phi_{l,j,i}(x_{l,i}) \tag{14}$$

We can simply understand that each neuron is connected to other neurons in the previous layer through a univariate function $\phi$. Similar to MLP, the computation of all neurons at layer $l$ can be reorganized as a function matrix multiplication $\mathbf{\Phi}_{l-1}$. Therefore, given a input vector $x \in \mathbb{R}^{n_0}$, the final output of KAN network is:

$$\text{KAN}(x) = (\mathbf{\Phi}_{L-1} \circ \cdots \circ \mathbf{\Phi}_1 \circ \mathbf{\Phi}_0)x \tag{15}$$

In vanilla KAN (Liu et al., 2024c), the univariate function $\phi_{l,j,i}$ is parametrized using B-splines, which is a class of smooth curves constructed via segmented polynomial basis functions. To ensure the stability and enhance the representational capacity, KAN overlays the spline function on a fixed basis function $b$, which is typically the SiLU function:

$$\phi(x) = w_b b(x) + w_s \text{spline(x)} \tag{16}$$

$$\text{spline}(x) = \sum_i c_i B_i(x) \tag{17}$$

where $w_b$ and $w_s$ are learnable weights and $\text{spline(x)}$ is the spline function constructed from the linear combination of B-spline basis functions $B_i$. However, the complex recursive computation process of high-order B-spline functions hinders the efficiency of KAN. Therefore, in this work, we adopt the simpler Chebyshev polynomial as the univariate function to replace the B-spline function (SS, 2024). The univariate function defined by the Chebyshev polynomial is given as follows:

$$T_k(x) = \cos(k \arccos(x)) \tag{18}$$

Here, $k$ represents the order of the polynomial. Then, we consider the univariate function $\Phi$ as a linear combination of Chebyshev polynomials with different orders:

$$x_{l+1,j} = \sum_{i=1}^{n_l} \phi_{l,j,i}(x_{l,i}) = \sum_{i=1}^{n_l} \sum_{k=0}^{K} \Theta_{i,k} T_k(\tanh(x_{l,i})) \tag{19}$$

Where $\Theta_{i,k}$ is the coefficients of $k$-th order Chebyshev polynomials acting on the $x_{l,i}$ and $\tanh$ is the tanh activation function used to normalize the inputs to between -1 and 1. By adjusting the highest order of the Chebyshev polynomial $K$, we can control the fitting capability of KAN. This also inspires tour design of the Multi-order KAN to dynamically represent different frequencies.

## B.2 FOURIER TRANSFORM

Time series are often composed of multiple frequency components superimposed on each other, and it is difficult to observe these individual frequency components directly in the time domain. Therefore, transforming a time series from the time domain to the frequency domain for analysis is often necessary. The Discrete Fourier Transform (DFT) is a commonly used domain transformation algorithm that converts a discrete-time signal from the time domain to the complex frequency domain. Mathematically, given a sequence of real numbers $x[n]$ in time domain, where $n = 0, 1, \ldots, N-1$ the DFT process can be described as:

$$X[k] = \sum_{n=0}^{N-1} x[n] \cdot e^{-i\frac{2\pi}{N}kn} = \sum_{n=0}^{N-1} x[n] \left( \cos\left(\frac{2\pi}{N}kn\right) - i\sin\left(\frac{2\pi}{N}kn\right) \right), \quad k = 0, 1, \ldots, N-1$$
(20)

where $X[k]$ is the $k$-th frequency component of frequency domain signal and $i$ is the imaginary unit. Similarly, we can use Inverse DFT (iDFT) to convert a frequency domain signal back to the time domain.

$$x[n] = \frac{1}{N} \sum_{k=0}^{N-1} X[k] \cdot e^{i\frac{2\pi}{N}kn} = \frac{1}{N} \sum_{k=0}^{N-1} X[k] \left( \cos\left(\frac{2\pi}{N}kn\right) + i\sin\left(\frac{2\pi}{N}kn\right) \right) \quad (21)$$

The computational complexity of the DFT is typically $\mathcal{O}(N^2)$(Zhou et al., 2022b). In practice, we use the Fast Fourier Transform (FFT) to efficiently compute the Discrete Fourier Transform (DFT) of complex sequences, which reduces the computational complexity to $\mathcal{O}(N \log N)$. Additionally, by employing the Real FFT (rFFT), we can compress an input sequence of $N$ real numbers into a signal sequence in the complex frequency domain containing $N/2 + 1$ frequency components.

