# OpenReview forum: "TimeKAN: KAN-based Frequency Decomposition Learning Architecture for Long-term Time Series Forecasting"
_ICLR.cc/2025/Conference — ICLR 2025 Poster_

### Official Review · Reviewer_BGvS · 2024-10-27

**Soundness:** 3
**Presentation:** 4
**Contribution:** 4
**Rating:** 8
**Confidence:** 5

**Summary:**

This paper explores a method for decomposing mixed frequency components into distinct single-frequency components to improve time series forecasting accuracy. The proposed approach, called TimeKAN, is based on the Kolmogorov-Arnold Network (KAN). TimeKAN's process consists of three key components: (1) Cascaded Frequency Decomposition (CFD) blocks, which use a bottom-up cascading approach to obtain series representations for each frequency band; (2) Multi-order KAN Representation Learning (M-KAN) blocks, which capture and represent specific temporal patterns within each frequency band; and (3) Frequency Mixing blocks, which recombine the separated frequency bands back into the original series format.

The study demonstrates that TimeKAN outperforms several state-of-the-art forecasting methods, including Autoformer, FEDformer, and iTransformer, by achieving lower MSE and MAE across multiple time series datasets such as Weather, ETTh2, and ETTm2.

**Strengths:**

1. Figure 1 is beautifully designed and provides an intuitive overview of each component in the new TimeKAN method, as well as how they connect.

2. The study makes effective use of large-scale datasets and performs comparisons with a variety of other methods (including CNN-based and Transformer-based models), demonstrating the advantages of TimeKAN. The model is also tested across different prediction lengths, and for datasets where performance is less optimal, the paper offers thorough explanations and detailed insights.

3. The analysis delves into several key components of TimeKAN, such as Upsampling, Depthwise Convolution, and Multi-order KANs. I especially appreciated this section, as it not only establishes that TimeKAN outperforms other deep learning methods but also shows that each individual component of TimeKAN is optimally designed.

**Weaknesses:**

1. Section 3.2 appears somewhat disorganized. While the overall logic is clear, the expression could be refined for clarity. Additionally, more mathematical details and background should be provided, which can be included in the appendix.

2. If possible, please add more data to Table 5 in Section 4.3. Supplement it with the performance of other methods in Table 1 on parameters (params) and MAC across these six datasets.

**Questions:**

N/A

---

> ### Author Response · Authors · 2024-11-22
> **Response to Reviewer BGvS**
>
> Many thanks to Reviewer BGvS for providing thorough detailed comments.
>
> > Q1: "Section 3.2 appears somewhat disorganized. While the overall logic is clear, the expression could be refined for clarity. Additionally, more mathematical details and background should be provided, which can be included in the appendix."
>
> Thank you for your valuable suggestions. We have reorganized and refined the content as follows:
> The sequence preprocessing part originally included in Section 3.2 Cascaded Frequency Decomposition has been relocated to a newly created section, Section 3.2 Hierarchical Sequence Preprocessing. We have also revised Section 3.3 Cascaded Frequency Decomposition to provide a clearer explanation of the step-by-step process used to obtain each frequency component in a cascaded manner. Furthermore, we have added detailed mathematical discussions on the Kolmogorov-Arnold Network and the Fourier Transform in Appendix A.
>
> > Q2: "If possible, please add more data to Table 5 in Section 4.3. Supplement it with the performance of other methods in Table 1 on parameters (params) and MAC across these six datasets."
>
>
>
> Thank you for your valuable suggestion. We have added the performance of the remaining methods from Table 1 in terms of parameters (params) and MACs across these six datasets. The complete comparison of parameters and MACs is shown below (Params\|MACs).
>
> | Datasets     | ETTH1           | ETTH2           | ETTm1           | ETTm2           | Weather          | Electricity     |
> | ------------ | --------------- | --------------- | --------------- | --------------- | ---------------- | --------------- |
> | TimeMixer    | 75.50K\|20.37M  | 75.50K\|20.37M  | 75.50K\|20.37M  | 77.77K\|24.18M  | 104.43K\|82.62M  | 106.83K\|1.26G  |
> | iTransformer | 841.57K\|77.46M | 224.22K\|19.86M | 224.22K\|19.86M | 224.22K\|19.86M | 4.83M\|1.16G     | 4.83M\|16.29G   |
> | PatchTST     | 3.75M\|5.90G    | 10.06M\|17.66G  | 3.75M\|5.90G    | 10.06M\|17.66G  | 6.90M\|35.30G    | 6.90M\|539.38G  |
> | TimesNet     | 605.48K\|18.13G | 1.19M\|36.28G   | 4.71M\|144G     | 1.19M\|36.28G   | 1.19M\|36.28G    | 150.30M\|4.61T  |
> | MICN         | 25.20M\|71.95G  | 25.20M\|71.95G  | 25.20M\|71.95G  | 25.20M\|71.95G  | 111.03K\|295.07M | 6.64M\|19.5G    |
> | Dlinear      | 18.62K\|0.6M    | 18.62K\|0.6M    | 18.62K\|0.6M    | 18.62K\|0.6M    | 18.62K\|0.6M     | 18.62K\|0.6M    |
> | FreTS        | 3.24M\|101.46M  | 3.24M\|101.46M  | 3.24M\|101.46M  | 3.24M\|101.46M  | 3.24M\|101.46M   | 3.24M\|101.46M  |
> | FILM         | 12.58M\|2.82G   | 12.58M\|2.82G   | 12.58M\|2.82G   | 12.58M\|2.82G   | 12.58M\|8.46G    | 12.58M\|8.46G   |
> | FEDFormer    | 23.38M\|24.96G  | 23.38M\|24.96G  | 23.38M\|24.96G  | 23.38M\|24.96G  | 23.45M\|25.23G   | 24.99M\|30.89G  |
> | AutoFormer   | 10.54M\|22.82G  | 10.54M\|22.82G  | 10.54M\|22.82G  | 10.54M\|22.82G  | 10.61M\|23.08G   | 12.14M\|28.75G  |
> | TimeKAN      | 12.84K\|7.63M   | 15.00K\|8.02M   | 14.38K\|7.63M   | 38.12K\|16.66M  | 20.94K\|29.86M   | 23.34K\|456.50M |
>
> As can be seen, except for DLinear, our TimeKAN consistently demonstrates a significant advantage in both parameter count and MACs compared to any other model. DLinear is a model consisting of only a single linear layer, which makes it the most lightweight in terms of parameters and MACs. However, the performance of DLinear already shows a significant gap when compared to state-of-the-art methods. In this case, the advantage in parameters may not be a major consideration for researchers. Therefore, in the main text, we compare the parameters and MACs of the current state-of-the-art methods: TimeKAN, TimeMixer, iTransformer, and PatchTST, and the complete comparison results are provided in Appendix B.2. Hope you can understand this arrangement.

---

> > ### Comment · Reviewer_BGvS · 2024-11-22
> >
> > Thank you for your response. I believe my questions have been resolved.

---

### Official Review · Reviewer_8Zzz · 2024-10-31

**Soundness:** 4
**Presentation:** 4
**Contribution:** 3
**Rating:** 8
**Confidence:** 2

**Summary:**

TimeKAN is a time series forecasting model that combines frequency decomposition, representation learning, and mixing. It first uses a moving average to separate high and low frequencies, creating multi-level sequences that are embedded into a high-dimensional space. Cascaded Frequency Decomposition (CFD) blocks progressively isolate each frequency band. The Multi-order KAN Representation Learning (M-KAN) blocks use Kolmogorov-Arnold Networks to capture temporal patterns within each frequency band independently. Finally, the Frequency Mixing blocks recombine these decomposed bands to restore the original sequence, which is then used for forecasting through a linear layer

**Strengths:**

1	Clarity and Structure: The paper follows a logical flow from problem statement to conclusions, making complex ideas accessible.
	2.	Thorough Background: A strong review of related work provides valuable context, situating the contribution within the field.
	3.	Detailed Experiments: Comprehensive experiments across multiple datasets support the model’s performance claims, with ablation studies highlighting component effectiveness.
	4.	Focused Writing: The paper stays on topic, avoiding unnecessary details and maintaining focus on the core contribution

**Weaknesses:**

Depth in Explanation: The methodology section could offer more detail on complex components like Kolmogorov-Arnold Networks for greater accessibility.

**Questions:**

What if we split the frequency band to more layers (more than 3 for example ). Will it increase performance ?

---

> ### Author Response · Authors · 2024-11-22
> **Response to Reviewer 8Zzz**
>
> We would like to sincerely thank Reviewer 8Zzz for providing a detailed review and insightful suggestions.
>
> > Q1: "Depth in Explanation: The methodology section could offer more detail on complex components like Kolmogorov-Arnold Networks for greater accessibility."
>
> Thanks to your valuable suggestions, we have added mathematical details about how Kolmogorov-Arnold Networks perform forward propagation as well as refined the description of the Chebyshev KAN in Section 3.4, please see the red text in Section 3.4.
>
> > Q2: "What if we split the frequency band to more layers (more than 3 for example ). Will it increase performance ?"
>
> In fact, the number of frequency bands is a hyperparameter that we manually set to accommodate the frequency domain characteristics of different datasets. To explore the impact of the number of frequency bands on performance, we set the number of frequency bands to 2, 3, 4, and 5. The effects of different frequency band divisions on performance (MSE\|MAE results under 96-to-96 setting) are shown in the table below:
>
> | Number of Frequency |      ETTh2       |     Weather      |   Electricity    |
> | :-----------------: | :--------------: | :--------------: | :--------------: |
> |          2          |   0.292\|0.340   |   0.164\|0.209   |   0.183\|0.270   |
> |          3          | **0.290\|0.339** |   0.163\|0.209   |   0.177\|0.268   |
> |          4          | **0.290\|0.340** | **0.162\|0.208** | **0.174\|0.266** |
> |          5          |   0.295\|0.346   |   0.164\|0.211   |   0.177\|0.273   |
>
> As we can see, in most cases, dividing the frequency bands into 3 or 4 layers yields the best performance. This aligns with our prior intuition: dividing into two bands results in excessive frequency overlap, while dividing into five bands leads to too little information within each band, making it difficult to accurately model the information within that frequency range. We have added the above discussion to Appendix B.5.

---

> > ### Comment · Reviewer_8Zzz · 2024-11-25
> > **split the frequency band**
> >
> > Could you please explain why splitting the band more than three times results in worse performance? Logically, doing so should result in better time and spectral resolution, which would enhance performance.

---

> ### Author Response · Authors · 2024-11-25
> **The Second Response to Reviewer 8Zzz**
>
> Logically,  more frequency bands could potentially improve performance by allowing for finer modeling. However, in real-life time series, noise is more prevalent in the high-frequency range. When the number of bands exceeds a certain threshold, some high-frequency bands may contain only useless noise, which harms prediction accuracy.
>
> One solution is to use an uneven frequency division, modeling only those bands with effective content. However, since the frequency distribution varies across different time series datasets, manually dividing the frequency for each specific dataset would be impractical and hinder model generalization. Therefore, we divide the frequency uniformly into multiple bands and limit the number of bands to a reasonable range (e.g., 3-4 bands), ensuring each band contains useful information for prediction while maintaining the model's generalization ability. In this way, we can apply band-specific modules to extract effective frequency data, improving the prediction performance. Our results show that 3 or 4 bands is an optimal threshold, beyond which the impact of noise on prediction is amplified, leading to a decline in performance.
>
> We hope our response addresses your question, and we would be happy to answer any further questions before the rebuttal ends.

---

### Official Review · Reviewer_gWX6 · 2024-11-04

**Soundness:** 3
**Presentation:** 3
**Contribution:** 2
**Rating:** 5
**Confidence:** 5

**Summary:**

In this paper, the authors propose a time series forecasting method based on KAN. TimeKAN uses frequency decomposition and KAN to effectively capture temporal correlations of the data. Experiments show the effectiveness of TimeKAN based on real-world datasets.

**Strengths:**

1. A vivid introduction and related work section to explain the background of time series forecasting and KAN.
2. A clear figure to illustrate the overall framework of TimeKAN. In the methodology section, all components are detailed.
3. Good ablation study to test the effectiveness of the different components of the model.

**Weaknesses:**

1. The author does not explain why TimeKAN does not perform well in the Electricity dataset. It would be helpful if the authors could provide potential reasons or hypotheses for why TimeKAN underperforms on the Electricity dataset specifically.
2. From Table 4, I do not see a huge increase with KAN compared to MLP models. Generally, if these results are similar, mostly, KAN is much slower than MLP. It would be good to see runtime comparisons between KAN and NLP implementations. Additionally, if KAN is slower than MLP in practice, it would be beneficial for authors to discuss more reasons why we prefer KAN over MLP.
3. For the look-back window, the authors do not compare TimeKAN with other models. For most models, when the prediction length is fixed, the prediction accuracy will increase as the look-back window increases. It is beneficial to provide a comparative analysis of TimeKAN's performance with varying look-back windows compared to other baseline models. This would provide a more comprehensive evaluation of TimeKAN's capabilities relative to existing methods.
4. For baseline methods, it is better to choose more frequency-based (such as FreTS) methods since frequency decomposition is a key contribution.

**Questions:**

1. See weaknesses.
2. Could you provide a short theoretical analysis about why in some cases in time series forecasting, KAN is better than MLP?
3. For Table 2, why not include the electricity dataset?
4. For KAN, you mentioned that the Kolmogorov-Arnold representation theorem states that any multivariate continuous function can be expressed as a combination of univariate functions and addition operations. Could you explain more about how it can capture multivariate correlations?

---

> ### Author Response · Authors · 2024-11-22
> **Response to Reviewer gWX6 [Part 1]**
>
> Many thanks to Reviewer gWX6 for providing thorough detailed comments.
>
> > Q1: "The author does not explain why TimeKAN does not perform well in the Electricity dataset. It would be helpful if the authors could provide potential reasons or hypotheses for why TimeKAN underperforms on the Electricity dataset specifically."
>
> Our TimeKAN is a model based on frequency analysis. **We infer that its poor performance on the electricity dataset is due to the overly short look-back window ($T=96$), which cannot provide sufficient frequency information.** To verify this, we compare the average number of effective frequency components under a specific look-back window. Specifically, we randomly select a sequence of length $T$ from the electricity dataset and transform it into the frequency domain using FFT. We define effective frequencies as those with amplitudes greater than 0.1 times the maximum amplitude. Then, we take the average number of effective frequencies obtained across all variables to reflect the amount of effective frequency information provided by the sequence. When $T=96$ (the setting in the paper), the average number of effective frequencies is 10.69. When we extend the sequence length to 512, the average number of effective frequencies becomes 19.74. Therefore, the effective frequency information provided by 512 time steps is nearly twice that of 96 time steps. This indicates that $T=96$ loses a substantial amount of effective information.
> To validate whether using $T=512$ allows us to leverage more frequency information, we extend the look-back window of TimeKAN to 512 on the electricity dataset and compare it with the state-of-the-art methods, TimeMixer and MOMENT. The results are shown in the table below (results of MSE\|MAE):
>
> | Models    |  **Predict-96**  | Predict-**192**  | **Predict-336**  | **Predict-720**  |
> | --------- | :--------------: | :--------------: | :--------------: | :--------------: |
> | MOMENT    |   0.136\|0.233   |   0.152\|0.247   |   0.167\|0.264   |   0.205\|0.295   |
> | TimeMixer |   0.135\|0.231   | **0.149\|0.245** |   0.172\|0.268   | **0.203**\|0.295 |
> | TimeKAN   | **0.133\|0.230** | **0.149**\|0.247 | **0.165\|0.261** | **0.203\|0.294** |
>
> Although TimeKAN performs significantly worse than TimeMixer when $T=96$, it achieves the best performance on the electricity dataset when the look-back window is extended to 512. This also demonstrates that TimeKAN can benefit significantly from richer frequency information. **For a fair comparison with other methods, we have adopted the common 96-step lookback window setting, which actually compromises the advantages of TimeKAN in long sequence.** We have added the above discussion to Appendix B.4.
>
>
>
> > Q2: "From Table 4, I do not see a huge increase with KAN compared to MLP models. Generally, if these results are similar, mostly, KAN is much slower than MLP. It would be good to see runtime comparisons between KAN and NLP implementations. Additionally, if KAN is slower than MLP in practice, it would be beneficial for authors to discuss more reasons why we prefer KAN over MLP."
>
> One significant reason why the improvement of KAN over MLP is not particularly pronounced lies in the well-designed nature of our Frequency Decomposition-Learning-Mixing architecture, which enables even MLP to demonstrate competitive performance. In TimeKAN, we use Chebyshev polynomials as the basis functions for the learnable activation functions, denoted as ChebyshevKAN. Assuming each layer of the neural network contains $N$ neurons, the computational cost per layer for ChebyshevKAN with a maximum degree of $K$ is $N^2K$, while for an MLP, the computational cost is $N^2$. Therefore, the computational cost of ChebyshevKAN is $K$ times that of an MLP.
> Notably, in the comparison presented in Table 4 of our paper, we ensure that the computational costs of the MLP and ChebyshevKAN are similar. Under comparable computational costs, we observe that the inference speeds (iter/s) of the two methods are not significantly different, as shown in the table below:
>
> | TimeKAN   | ETTh1   | ETTm1   | Weather |
> | --------- | ------- | ------- | ------- |
> | MLP-based | 0.0124s | 0.0148s | 0.0246s |
> | KAN-based | 0.0121s | 0.0159s | 0.0250s |
>
> Therefore, under similar computation consumption, KAN achieves relatively better performance, demonstrating its capability in dynamic frequency representation. With the emergence of more KAN variants, we can anticipate further improvements in the computational efficiency and forecasting performance of TimeKAN.

---

> ### Author Response · Authors · 2024-11-22
> **Response to Reviewer gWX6 [Part 2]**
>
> > Q3: "Lack of comparison between TimeKAN and other methods regarding look-back windows."
>
> Thank you for your valuable suggestion. We have added the performance of TimeMixer, iTransformer and PatchTST with varing look back windows for comparison in Figure 2. You can also see the table ( MSE results of 96 step forecasting horizon in Ettm2 dataset) below.
>
> | Models       | Iuput-48  | Iuput-96  | Iuput-192 | Iuput-336 | Iuput-512 | Iuput-720 |
> | ------------ | --------- | --------- | --------- | --------- | --------- | --------- |
> | PatchTST     | 0.191     | 0.183     | 0.176     | 0.173     | 0.173     | 0.188     |
> | iTransformer | 0.193     | 0.180     | 0.177     | 0.175     | 0.179     | 0.181     |
> | TimeMixer    | **0.187** | 0.175     | 0.171     | 0.179     | 0.170     | 0.176     |
> | TimeKAN      | 0.190     | **0.174** | **0.170** | **0.167** | **0.163** | **0.164** |
>
> As we have seen, TimeKAN can clearly benefit from longer look back windows compared to other models and consistently performs optimally in most cases.
>
>
>
> > Q4: "For baseline methods, it is better to choose more frequency-based (such as FreTS) methods since frequency decomposition is a key contribution."
>
> Thanks to your valuable suggestions, we have added two representative frequency methods FiLM and FreTS in Table 1. A comparison between TimeKAN and these two frequency methods is shown in the following table (Average MSE|MAE of all forecasting horizons):
>
> |              |                   |                   |                   |                   |                   |                   |
> |:------------:|:-----------------:|:-----------------:|:-----------------:|:-----------------:|:-----------------:|:-----------------:|
> | **Datasets** | **ETTh1**         | **ETTh2**         | **ETTm1**         | **ETTm2**         | **Weather**       | **Electricity**   |
> | Metric       | MSE\|MAE          | MSE\|MAE          | MSE\|MAE          | MSE\|MAE          | MSE\|MAE          | MSE\|MAE          |
> | FreTS        | 0.491\| 0.475     | 0.433\| 0.446     | 0.407\| 0.417     | 0.339\| 0.374     | 0.245\| 0.294     | **0.192\| 0.282** |
> | FiLM         | 0.516\| 0.483     | 0.402\| 0.420     | 0.412\| 0.402     | 0.288\| 0.328     | 0.271\| 0.290     | 0.223\| 0.302     |
> | TimeKAN      | **0.417\| 0.427** | **0.383\| 0.404** | **0.376\| 0.395** | **0.277\| 0.322** | **0.242\| 0.272** | 0.197\| 0.286     |
>
>
>
> It can be seen that TimeKAN significantly outperforms FiLM and FreTS on most of the datasets, which shows the effectiveness of our designed frequency Decomposition-Learning-Mixing architecture.

---

> ### Author Response · Authors · 2024-11-22
> **Response to Reviewer gWX6 [Part 3]**
>
> > Q5: "Could you provide a short theoretical analysis about why in some cases in time series forecasting, KAN is better than MLP?"
>
> Time series typically consist of components from multiple frequencies. Suppose we divide the time series into $K$ frequency bands from low to high frequency, where each frequency band $f_k$ contains $N$ frequencies. The complexity $C_{f_k}$ of the $k$-th frequency band $f_k$ can be defined as its spectral entropy, i.e., $C_{f_k}=-\sum_{i=1}^N P(f_i) \log P(f_i)$
> where $P(f_i)$ is the normalized power spectrum. In most cases, the time series contains a small number of concentrated high-amplitude frequencies in the low-frequency region, resulting in lower complexity. On the other hand, the distribution in the high-frequency region is relatively uniform, leading to higher complexity. Therefore, the complexity of each frequency band usually satisfies $C_{f_1} < C_{f_2} < \cdots < C_{f_K}$.
> To perform more accurate time series prediction, we assume that for each frequency band $f_k$, its representation in the latent space is $F_k$. Furthermore, assume there exists an perfect network $M_k$ that perfectly fits $F_k$ in the latent space, such that $M_k(f_k) = F_k.$
> Thus, the complexity of the network should be proportional to the complexity of the corresponding frequency band, satisfying $C_{M_1} < C_{M_2} < \cdots < C_{M_K}$. Our goal is to find the network $\hat{M}\_k$ such that the following fitting error is minimized: $(\hat{M}\_k(f_k) - M_k(f_k))$ for each $f_{k}$. To achieve this, we first need to minimize the complexity error betweem $\hat{M}\_k(f_k)$ and $M_k(f_k)$.  We define the complexity error for the $k$-th frequency band as $\Delta C_k = C_{\hat{M}\_k} - C_{M_k}$, where $C_{\hat{M}\_k}$ represents the complexity of the fitted network $\hat{M}\_k$. When using  MLP as the fitting network, the complexity $C_{\hat{M}\_k}$ is fixed. Since the complexity of the MLP cannot be adjusted, there can be at most one $C_{\hat{M}\_k}$ that satisfies:
> $\Delta C_k = 0$, because the  $C_{M_k}$ is monotonically increasing with $k$. Therefore, when using the fixed-complexity MLP as the fitting network, there will inevitably be an error accumulation, and the total error satisfies $\sum_{k=1}^K \Delta C_k \geq \epsilon$, where $\epsilon$ is a lower bound for the error. In contrast, when using KAN, we can flexibly control the complexity by adjusting the polynomial degree of the KAN's internal kernel. In the optimal case, each frequency band can satisfy $\Delta C_k = 0$, resulting in a total error of zero: $\sum_{k=1}^K \Delta C_k = 0$. This shows that the KAN can achieve a better theoretical performance upper bound when fitting multiple frequency components in the latent space. Furthermore, by accurately fitting the frequency representations, KAN can provide more reliable foundations for precise time series prediction.

---

> ### Author Response · Authors · 2024-11-22
> **Response to Reviewer gWX6 [Part 4]**
>
> > Q6: "For Table 2, why not include the electricity dataset?"
>
> Thanks to your valuable reminder, we have added the results of the missing electricity dataset in Table 2, which is shown in the following table:
>
> |                          |                  |                  |                  |                  |                  |                  |
> |--------------------------|:----------------:|:----------------:|:----------------:|:----------------:|:----------------:|:----------------:|
> | **Datasets**             | **ETTh1**        | **ETTh2**        | **ETTm1**        | **ETTm2**        | **Weather**      | **Electricity**  |
> | Metric                   | MSE\|MAE         | MSE\|MAE         | MSE\|MAE         | MSE\|MAE         | MSE\|MAE         | MSE\|MAE         |
> | Linear Mapping           | 0.401\|0.413     | 0.312\|0.362     | 0.328\|0.365     | 0.180\|0.263     | 0.164\|0.211     | 0.184\|0.275     |
> | Linear Interpolation     | 0.383\|0.398     | 0.296\|0.347     | 0.336\|0.370     | 0.181\|0.263     | 0.165\|0.210     | 0.196\|0.277     |
> | Transposed Convolution   | 0.377\|0.407     | **0.290\|0.344** | 0.326\|0.366     | 0.178\|0.261     | 0.163\|0.211     | 0.188\|0.274     |
> | **Frequency Upsampling** | **0.367\|0.395** | **0.290\|0.340** | **0.322\|0.361** | **0.174\|0.255** | **0.162\|0.208** | **0.174\|0.266** |
>
>
>
>
> >  Q7: "For KAN, you mentioned that the Kolmogorov-Arnold representation theorem states that any multivariate continuous function can be expressed as a combination of univariate functions and addition operations. Could you explain more about how it can capture multivariate correlations?"
>
> When using KAN to perform supervised learning tasks, the correlations among multivariate are implicitly reflected in the learnable univariate activation functions. Each variable in KAN is regulated by a learnable univariate activation function. If multiple variables exhibit consistent behavior, KAN tends to reduce redundancy by allowing only a small number of activation functions to remain effective, while deactivating those corresponding to redundant variables. Through supervised training, KAN learns and captures the implicit correlations among variables and uses the learnable activation functions to regulate their behavior.
> If explicit direct relationships among variables are needed, unsupervised learning can be employed to find a non-zero function $g$ such that $g(x_{1}, \cdots, x_{n})\approx0$. After training, the explicit relationships between each variable and the remaining variables can be constructed by extracting the expressions of the corresponding learnable activation functions.
>  In this scenario, the univariate refers to one element in the input sequence and does not represent a variate in the multivariate time series. In TimeKAN, KAN is used to model the latent representation of time series, i.e., learning the hidden dimension representation.  And we adopt a variate-independent strategy for multivariate processing, also known as the channel-independent strategy in PatchTST.

---

> > ### Comment · Reviewer_gWX6 · 2024-11-23
> >
> > Thank you for your response, which has addressed my concerns to a certain extent. However, after reading other reviews, I share Reviewer XpEF's concerns regarding the validity of KAN itself. Therefore, I chose to maintain my original score, but I will discuss this further with the AC and other reviewers in the next stage to decide whether to change the score.

---

> > > ### Author Response · Authors · 2024-12-03
> > > **Gentle Reminder**
> > >
> > > Dear Reviewer gWX6,
> > >
> > > We would like to sincerely thank you once again for your valuable feedback. We fully understand your concerns about the effectiveness of KAN as a novel neural network. However, with less than 24 hours remaining in the rebuttal period, we would like to kindly remind you that vanilla KAN has already received all positive evaluations from five reviewers on OpenReview, and its theoretical contributions have been recognized by the community. In TimeKAN, we have thoughtfully designed the Multi-order KAN to better align the variable characteristics of KAN with the multi-frequency nature of time series, and we have demonstrated its effectiveness through experimental results. Given this, we genuinely hope that you will reconsider your score. Finally, we would like to express our sincere appreciation for your constructive comments, which have greatly enhanced the quality of our paper.
> > >
> > > Best regards,
> > >
> > > The Authors

---

### Official Review · Reviewer_XpEF · 2024-11-07

**Soundness:** 1
**Presentation:** 2
**Contribution:** 1
**Rating:** 3
**Confidence:** 5

**Summary:**

The paper presents TimeKAN, a Kolmogorov-Arnold Network (KAN)-based model for long-term time series forecasting, designed to handle complex multi-frequency patterns in real-world data. Traditional models struggle with mixed frequencies, but TimeKAN addresses this with a three-part architecture: Cascaded Frequency Decomposition to separate frequency bands, Multi-order KAN Representation Learning to model each band’s specific patterns using adaptable polynomial orders, and Frequency Mixing to recombine frequencies effectively. Experiments show that TimeKAN achieves superior accuracy and efficiency compared to existing models, making it a robust, lightweight solution for complex TSF tasks.

**Strengths:**

**Exploratory Application of KAN to Time Series Forecasting**: The paper attempts to introduce Kolmogorov-Arnold Networks (KAN) into time series forecasting, using multi-order polynomial representations to handle the complexities of different frequency components. While KAN has not been widely applied in this area, this effort demonstrates its potential flexibility in data fitting and offers an alternative approach to traditional MLPs.

 **Comprehensive Experimental Design**: The paper includes experiments across various time series datasets, such as weather, electricity, and energy data, covering diverse scenarios. Additionally, it conducts ablation studies to examine the effects of each module. These experiments help to assess TimeKAN’s performance and may provide a reference point for further research.

**Weaknesses:**

**Lack of Innovation**: The primary contribution of this paper is the integration of the Kolmogorov-Arnold Network (KAN) into time series forecasting, yet the work does not introduce novel methods or substantial breakthroughs in methodology. While the inclusion of KAN is somewhat new, other components, such as frequency decomposition and mixing, are mature techniques, and the paper does not propose innovative applications or enhancements to these. Overall, this work appears more like a combination of existing technologies rather than a genuinely innovative study.

**Absence of Comparison with Cutting-Edge Models**: The experiments lack direct comparisons with state-of-the-art models, especially those in high demand for time series forecasting, such as large language models (LLMs) and foundation models. Given current research trends, these models have become widely adopted benchmarks. Without such comparisons, the effectiveness of the proposed method remains unclear, especially as the improvements presented are relatively limited compared to advancements in mainstream approaches.

 **Reliance on KAN, a Model with Limited Validation**: The foundation of TimeKAN is the KAN model, which has not yet been widely validated or accepted. Its theoretical correctness and practical effectiveness remain uncertain, which casts doubt on the reliability and generalizability of TimeKAN as a whole. If there are inherent issues with KAN, the predictive performance and stability of TimeKAN could be compromised, making the paper's conclusions less convincing.

**Insufficient Analysis of Computational Efficiency**: While the paper claims that TimeKAN is more lightweight than existing methods, it lacks an in-depth analysis of its actual computational efficiency, especially compared to more mainstream and optimized time series models. Additionally, there is no quantification of the computational cost associated with KAN’s multi-order polynomial calculations when handling long-sequence data. Given that many time series tasks require efficient real-time computations, focusing solely on parameter reduction does not adequately demonstrate TimeKAN’s advantage in computational efficiency; the absence of data on inference speed and computational cost undermines its practical applicability.


**Focus on Single-Task Performance Rather Than Generalized Representation**: A dominant trend in time series modeling now follows the approach of large language models (LLMs) to develop foundation models and leverage self-supervised representation learning. This approach enables generalization across various tasks and domains, ultimately aiming for a “one-fit-all” solution. However, this paper remains focused on improving single-task performance in time series forecasting (TSF), which may be of limited value in light of the broader goals of the field. Furthermore, the improvements reported in the experimental results are relatively modest, and without statistical significance testing, it remains unclear if these gains are truly meaningful or could simply be attributed to random variation.

**Questions:**

**To my knowledge, KAN itself has not been formally accepted, meaning it has not undergone rigorous peer-reviewed validation. If KAN’s theoretical foundation is later found to be flawed, would this impact the validity of this paper?** – If the underlying theory or structure of KAN is later shown to have limitations or inaccuracies, would the overall reliability of TimeKAN be compromised? Has the author considered this risk, and are there alternative solutions in place?

**In the experimental section, this paper does not compare TimeKAN with current state-of-the-art models (such as large language models or foundation models), making it difficult to assess its actual performance** – Without direct comparisons with these advanced models, can TimeKAN demonstrate a significant advantage? If the authors believe TimeKAN holds particular value in computational efficiency or predictive accuracy, could more data be provided to quantify this advantage?

**A major trend in time series research is developing foundation models, inspired by large language models, to generalize across domains and tasks. However, it is unclear if TimeKAN’s current design can support such robust representation learning** – Can TimeKAN truly compete with established frameworks like Transformers in terms of generalization and adaptability? If not, have the authors considered alternative approaches to enhance TimeKAN’s structural robustness and flexibility for broader applications?

---

> ### Author Response · Authors · 2024-11-22
> **Response to Reviewer XpEF [Part1]**
>
> We would like to sincerely thank Reviewer XpEF for providing the valuable feedback.
>
> >**Q1:** "This work appears more like a combination of existing technologies, such as frequency decomposition and mixing, rather than a genuinely innovative study."
>
> In fact, the core contribution of TimeKAN lies in re-examining the time series forecasting task from the perspective of multi-frequency decomposition learning.  **To the best of our knowledge, TimeKAN is the first method to decompose time series into multiple frequency bands represented in the time domain and leverage time-domain processing techniques to learn representations for specific frequencies, which effectively handle the challenges arising from complex frequency mixing.** As a result, TimeKAN fundamentally differs from prior methods in frequency domain learning (e.g., FreTS, FEDformer) and decomposition-based learning (e.g., TimeMixer, MICN) in the following ways:
>
> (1) Frequency-Domain Learning: FreTS, FEDformer, and other frequency-domain based methods focus on learning the representation of frequency signals directly in the frequency domain. However, current evidence suggests that methods learning directly in the frequency domain generally perform worse than those learning directly in the time domain. In contrast, we propose using Frequency Upsampling to cascade different frequency components’ representations in the time domain, which allows TimeKAN to directly process different frequency components in the time domain, avoiding the difficulty of learning frequency domain signals directly. To our knowledge, no previous work has adopted this paradigm.
>
> (2) Decomposition-Based Learning: Decomposition-based methods like TimeMixer and MICN mostly decompose sequences into subcomponents from a time-domain perspective, while we decompose time series from a frequency-domain perspective. Frequency domain perspective enables a more comprehensive analysis of time series, encompassing both periodic long-term variations and sharp short-term fluctuations. By extracting the sequence from the frequency domain and learning in the time domain, our TimeKAN is able to accurately model the subcomponents of the time series.
>
> Furthermore, the incorporation of the Multi-order KAN enhances the framework’s capability  to adaptively learn specific frequency components. TimeKAN has been demonstrated state-of-the-art performance across multiple long-term time series forecasting tasks and showcasing its lightweight and parameter-efficient architecture.

---

> ### Author Response · Authors · 2024-11-22
> **Response to Reviewer XpEF [Part 2]**
>
> > Q2: "Lack of comparison between TimeKAN and current state-of-the-art LLM-based time series models as well as time series foundation models in terms of predictive accuracy or computational efficiency."
>
> Thanks for your valuable suggestion. **Through comparisons, our TimeKAN outperforms the state-of-the-art LLM-based time series model and time series foundation model in the majority of cases across six datasets.** We have added a state-of-the-art LLM-based time series model TimeFFM [1], which have the same look-back window ($T=96$) setting as TimeKAN. Additional comparative results are shown below:
>
> | Models             | Predict |   ETTh1   |   ETTh2   |   ETTm1   |   ETTm2   |  Weather  | Electricity |
> | ------------------ | ------- | :-------: | :-------: | :-------: | :-------: | :-------: | :---------: |
> | **TimeFFM (T=96)** | 96      |   0.385   |   0.301   |   0.336   |   0.181   |   0.191   |    0.198    |
> |                    | 192     |   0.439   |   0.378   |   0.378   |   0.247   |   0.236   |    0.199    |
> |                    | 336     |   0.480   | **0.422** |   0.411   |   0.309   |   0.289   |    0.212    |
> |                    | 720     |   0.462   | **0.427** |   0.469   |   0.406   |   0.362   |    0.253    |
> | **TimeKAN (T=96)** | 96      | **0.367** | **0.290** | **0.322** | **0.174** | **0.162** |  **0.174**  |
> |                    | 192     | **0.414** | **0.375** | **0.357** | **0.239** | **0.207** |  **0.182**  |
> |                    | 336     | **0.445** |   0.423   | **0.382** | **0.301** | **0.263** |  **0.197**  |
> |                    | 720     | **0.444** |   0.443   | **0.445** | **0.395** | **0.338** |  **0.236**  |
>
> Additionally, we also compare our TimeKAN with the state-of-the-art time series foundation model MOMENT [2], which is pretrained on time series dataset of size 13M. Notably, the look back window of MOMENT is set to 512. Therefore, we extend the look window of TimeKAN to 512 to ensure a fair comparison. The result of the comparison is as follows:
>
> | Models              | F    |   ETTh1   |   ETTh2   |   ETTm1   |   ETTm2   |  Weather  | Electricity |
> | ------------------- | ---- | :-------: | :-------: | :-------: | :-------: | :-------: | :---------: |
> | **MOMENT (T=512)**  | 96   |   0.387   |   0.288   | **0.293** |   0.170   |   0.154   |    0.136    |
> |                     | 192  |   0.410   |   0.349   |   0.326   |   0.227   |   0.197   |    0.152    |
> |                     | 336  |   0.422   |   0.369   | **0.352** |   0.275   |   0.246   |    0.167    |
> |                     | 720  |   0.454   |   0.403   |   0.405   |   0.363   |   0.315   |    0.205    |
> | **TimeKAN (T=512)** | 96   | **0.357** | **0.266** | **0.293** | **0.163** | **0.143** |  **0.133**  |
> |                     | 192  | **0.401** | **0.138** | **0.323** | **0.224** | **0.186** |  **0.149**  |
> |                     | 336  | **0.408** | **0.333** |   0.353   | **0.274** | **0.237** |  **0.165**  |
> |                     | 720  | **0.447** | **0.376** | **0.398** | **0.360** | **0.309** |  **0.203**  |
>
> As shown in the two tables above, our TimeKAN outperforms the state-of-the-art LLM for time series models and time series foundation models in the majority of cases across six datasets, demonstrating its effectiveness in time series forecasting. Furthermore, Time-FFM has 90M parameters, and MOMENT has 385M parameters, while TimeKAN has an average of only 21K parameters. Therefore, TimeKAN is a simple yet effective model, achieving superior performance with just 0.025\% (or even less) of the parameters required by these foundational models.
>
> [1] Liu et al., Time-FFM: Towards LM-Empowered Federated Foundation Model for Time Series Forecasting.,  NeurIPS 2024.
>
> [2]Goswami et al., MOMENT: A Family of Open Time-series Foundation Models., ICML 2024.

---

> ### Author Response · Authors · 2024-11-22
> **Response to Reviewer XpEF [Part 3]**
>
> > Q3: "The foundation of TimeKAN is KAN model, which has limited validation and uncertain theoretical correctness. If KAN is found to have flaws, it could undermine overall reliability of TimeKAN."
>
> In fact, **the foundation of TimeKAN is Decomposition-Learning Mixing architecture, which decompose time series into multiple frequency bands represented in the time domain and leverage time-domain processing techniques to learn representations for specific frequencies.** We have demonstrated the effectiveness of this framework, showing that even when KAN is replaced with a traditional MLP, competitive performance is still maintained. To better represent various frequency components in a more fitting manner, we further developed the Multi-order KAN to enhance the Decomposition-Learning Mixing architecture. In addition, based on the Decomposition-Learning Mixing architecture, the performance of the well-designed Multi-order KAN surpasses that of the direct use of KAN, indicating that a carefully designed Multi-order KAN can more accurately model different frequency components. As better KAN variants emerge in the future, our framework can benefit further. Therefore, from a practical perspective, our current efforts have shown that the design of the Multi-order KAN is an effective component of our framework.
>
> From a theoretical standpoint, although KAN is still a relatively new member of the deep learning family, the Kolmogorov-Arnold Network theorem it relies on has been established for decades. The Kolmogorov-Arnold Network theorem can be described as: any multivariate continuous function can be represented as a combination of univariate functions and addition operations. In our Multi-order KAN, we primarily utilize univariate functions of varying complexities to form networks with different fitting capabilities, thus representing different frequency components, which is essentially the core concept of the Kolmogorov-Arnold Network theorem. Therefore, we believe that the use of Multi-order KAN can be fundamentally attributed to the well-established Kolmogorov-Arnold Network theorem, making the design of our TimeKAN consistent with foundational theory.
>
> > Q4-part1: "Lack of in-depth analysis of the actual computational efficiency between TimeKAN and mainstream time series models."
>
> Thank you for your valuable feedback. We have added an analysis of the computational complexity of TimeKAN in Appendix B.1. Given a multivariate time series with $M$  variables and  $L$ time steps, the computational complexity of TimeKAN is approximately $\mathcal{O}(ML\ logL)$. This indicates that the complexity scales linearly with the number of variables and grows logarithmically with the number of time steps.  The term $L\ logL$ arises from the computational complexity of the Fast Fourier Transform. To further reduce the computational complexity with respect to the number of time steps, we can employ a pre-selected set of Fourier bases for quick implementation, reducing the complexity to $\mathcal(ML)$.  In comparison, the computational complexity of PatchTST is $\mathcal(L^{2})$, iTransformer has a complexity of $\mathcal(M^{2})$, and TimeMixer exhibits a complexity of $ML^{2}$.Therefore, theoretically, TimeKAN achieves a better balance between the number of variables and input length when compared to these state-of-the-art methods. Notably, the actual computational speed may vary due to differences in parameter settings. For instance, while the computational complexity of PatchTST scales quadratically with input length, its computation speed can be significantly improved by selecting larger patch sizes.
>
> > Q4-part2: "There is no quantification of the computational cost associated with KAN’s multi-order polynomial calculations when handling long-sequence data. "
>
> In TimeKAN, the role of the Multi-order KAN is to learn representations at different frequencies, i.e., learning the latent space representation. For a single KAN, assuming the highest degree of the Chebyshev polynomial is $k$ and the hidden dimension is $D$, the computational cost for channel modeling is $M\*L\*K\*D^{2}$, which increases linearly with the sequence length $L$. Therefore, it does not result in excessive computational costs for long sequences.

---

> ### Author Response · Authors · 2024-11-22
> **Response to Reviewer XpEF [Part 4]**
>
> > Q4-part3: "The absence of data on inference speed and computational cost undermines its practical applicability."
>
> In fact, the computational cost of TimeKAN has been presented in Table 5 using the MACs metric. Compared with the three other best-performing models (including MLP-based and Transformer-based methods), TimeKAN requires significantly less computational cost. For inference time (s/iter), we have added a comparison of TimeKAN and other models under the same settings. Due to the computational cost of multiple Fourier Transforms, TimeKAN’s inference speed is relatively slower but remains in the same order of magnitude as other methods except time series foundation model Time-FFM, and significantly outperforms Time-FFM.
>
> | Models       | ETTh1   | ETTm1   | Weather |
> | ------------ | ------- | ------- | ------- |
> | Time-FFM     | 0.0216s | 0.0219s | 0.0539s |
> | PatchTST     | 0.0107s | 0.0116s | 0.0208s |
> | TimeMixer    | 0.0136s | 0.0155s | 0.0223s |
> | iTransformer | 0.0114s | 0.0126s | 0.0185s |
> | TimeKAN      | 0.0121s | 0.0159s | 0.0250s |
>
>
>
> > Q5-part1: "Compared to one-fit-all solutions like LLM-based time series methods or time series foundation methods, approaches focusing solely on improving time series forecasting may have limited value. Besides, TimeKAN may not be able to compete with established frameworks like Transformers in terms of generalization and adaptability."
>
> Our TimeKAN is carefully designed for time series forecasting. As shown in the table above, TimeKAN significantly outperforms the current state-of-the-art LLM-based time series methods Time-FFM and time series foundation models MOMENT. Furthermore, as recent studies indicate: LLM-based time series forecasters perform the same or worse than basic LLM-free ablations, yet require orders of magnitude more compute [3]. This suggests that the effectiveness of current LLMs for the time series forecasting task remains questionable. Besides, the state-of-the-art time series foundation model MOMENT requires pretraining on over 13 million time series datasets, which consumes approximately 404 GPU hours and has more than 20000 times the number of parameters as TimeKAN. However, it still fails to outperform TimeKAN on typical time series forecasting tasks. Therefore, whether achieving superior performance in time series forecasting requires building large models following trends in other fields or designing lightweight, task-specific models remains an open question.
>
> **Based on the current evidence, large models trained on massive datasets have not surpassed our lightweight TimeKAN in time series forecasting tasks.** While such models perform well in generalizing across multiple tasks, they are yet to achieve the best results on single forecasting task. In conclusion, we argue that the future of time series forecasting should not be confined to scaling up models. Designing high-performance, lightweight models that incorporate domain knowledge for single tasks presents a promising alternative pathway.
>
> > Q5-part2: "The improvements reported in the experimental results are relatively modest, and without statistical significance testing, it remains unclear if these gains are truly meaningful or could simply be attributed to random variation. "
>
> To evaluate the robustness of TimeKAN, we repeated the experiments on three randomly selected seeds and compared it with the second-best model (TimeMixer). We report the mean and standard deviation of the results across the three experiments, as well as the confidence level of TimeKAN's superiority over TimeMixer. The results are averaged over four prediction horizons (96, 192, 336, and 720).
>
> | Dataset |                 |   **MSE**   |            |      |                 |   **MAE**   |            |
> | ------- | :-------------: | :---------: | :--------: | ---- | :-------------: | :---------: | :--------: |
> |         |     TimeKAN     |  TimeMixer  | Confidence |      |     TimeKAN     |  TimeMixer  | Confidence |
> | ETTh1   | **0.422±0.004** | 0.462±0.006 |    99%     |      | **0.430±0.002** | 0.448±0.004 |    99%     |
> | ETTh2   | **0.387±0.003** | 0.392±0.003 |    99%     |      | **0.408±0.003** | 0.412±0.004 |    90%     |
> | ETTm1   | **0.378±0.002** | 0.386±0.003 |    99%     |      | **0.396±0.001** | 0.399±0.001 |    99%     |
> | ETTm2   |   0.278±0.001   | 0.278±0.001 |     —      |      | **0.324±0.001** | 0.325±0.001 |    90%     |
> | Weather | **0.243±0.001** | 0.245±0.001 |    99%     |      | **0.273±0.001** | 0.276±0.001 |    99%     |
>
> As shown in the table, in most cases, we have over 90% confidence that TimeKAN outperforms the second-best model and demonstrates good robustne of TimeKAN.
>
> [3] Tan et al., Are Language Models Actually Useful for Time Series Forecasting?,  NeurIPS 2024.

---

> > ### Comment · Reviewer_XpEF · 2024-11-22
> >
> > I appreciate that the author has responded to all the questions I raised in a very serious and comprehensive manner. However, since this paper is based on the KAN algorithm, from my personal experience in using the KAN algorithm in the past, I do have some concerns and doubts that are hard to let go regarding its performance and efficiency. At this stage, before the KAN algorithm has successfully passed the strict peer review process and been finally accepted, and also before it has obtained widespread recognition and affirmation in the academic community, I think this paper may not be able to meet the acceptance criteria for the time being. Therefore, I have decided to maintain the current score evaluation.

---

> ### Author Response · Authors · 2024-12-03
> **Kind Reminder**
>
> Dear Reviewer XpEF,
> ﻿
>
> We would like to express our sincere gratitude once again for your valuable feedback. We fully understand your concerns about the effectiveness of KAN as a novel neural network, but with less than 24 hours remaining until the rebuttal period concludes, we would like to kindly remind you that the vanilla KAN has already received all positive evaluations from five reviewers on OpenReview, and its theoretical contributions have been acknowledged by the community. This demonstrates the potential of KAN for widespread adoption in the deep learning community. In TimeKAN,  we have carefully designed the Multi-order KAN to align the variable characteristics of KAN with the multi-frequency nature of time series, and demonstrated its effectiveness through experiments. Therefore, we sincerely hope that you might reconsider your score. Finally, we would like to reiterate our appreciation for your thoughtful comments, which have significantly improved the quality of our paper.
> ﻿
>
> Best regards,
> ﻿
>
> The Authors

---

### Author Response · Authors · 2024-11-25

We sincerely appreciate all the reviewers for their thoughtful feedback and constructive suggestions, which have been immensely helpful in guiding us to enhance our paper. We would like to clarify three points in order to address the remaining concerns of some reviewers to the best of our ability:

(1)**The core foundation of TimeKAN lies in the Decomposition-Learning-Mixing architecture rather than KAN**, which decomposes time series into multiple frequency bands represented in the time domain and leverages time-domain processing techniques to learn representations for specific frequencies.  Multi-order KAN are carefully designed to efficiently adapt this framework and improve forecasting performance across multiple datasets. TimeKAN has been demonstrated state-of-the-art performance across multiple long-term time series forecasting tasks and showcasing its lightweight and parameter-efficient architecture.

(2)We understand the the concerns of some reviewers regarding the effectiveness of KAN. However, **we emphasize that KAN is not a seamless replacement for MLP and may not work in all cases. It requires careful design to  fully realize its potential.** As shown in Table 4 of the paper, directly replacing MLP with KAN results in a performance that is actually worse than MLP. Therefore, in TimeKAN, we have designed the Multi-order KAN to better match the modeling of different frequency components, thereby maximizing the advantages of KAN’s variable complexity. And the results demonstrate that Multi-order KAN outperforms both the direct use of KAN and the direct use of MLP.

(3) To the best of our knowledge, **the vanilla KAN has received positive feedback on OpenReview, with five reviewers consistently agreeing on its contribution to the community, which to some extent demonstrates its effectiveness.** Therefore, we hope the reviewers will approach KAN with a more open perspective.

We sincerely hope the reviewers will reconsider their score and look forward to further feedback.

---

### Meta-Review · Area_Chair_cbq9 · 2024-12-21

**Metareview:**

The paper proposes a novel forecasting method that exploits the frequency structure of time series. The method, TimeKAN, has a decomposition component, a representations learning component based on the Kolmogorov-Arnold Network, and a mixing component.

Overall, this paper has a solid technical contribution and convincing experiments. The reviewers also praised the clarity of the paper, the thorough presentation of background work, and the ablation studies.

There were some reservations expressed because the KAN algorithm has not yet been accepted at a conference - however, it has been public on arXiv since April 2024 and the proposed method is sufficiently different from that algorithm, so I do not see it as a problem, particularly since the authors have conducted additional experiments replacing the KAN structure with an MLP, showing the merits of the other components in their architecture (i.e. the method is not exclusively reliant on the success of KAN).

The concerns raised by reviewer XpEF about comparisons to state-of-the-art models have been adequately resolved in the rebuttal through the addition of comparisons to moment and TimeFFM.

Thus, I recommend acceptance of this submission.

**Additional Comments On Reviewer Discussion:**

As mentioned in the meta-review above, there were some concerns expressed with respect to the KAN-based block in the paper and comparisons to s.o.t.a. which, in my view, were addressed by the authors.

---

### Decision · Program_Chairs · 2025-01-22

Accept (Poster)